# Joint Attribute and Model Generalization Learning for Privacy-Preserving Action Recognition

**Duo Peng**
SUTD
Singapore
duo_peng@mymail.sutd.edu.sg

**Li Xu**
SUTD
Singapore
li_xu@mymail.sutd.edu.sg

**Qiuhong Ke**
Monash University
Australia
Qiuhong.Ke@monash.edu

**Ping Hu**
UESTC
China
chinahuping@gmail.com

**Jun Liu**[*]
SUTD
Singapore
jun_liu@sutd.edu.sg

## Abstract

Privacy-Preserving Action Recognition (PPAR) aims to transform raw videos into anonymous ones to prevent privacy leakage while maintaining action clues, which is an increasingly important problem in intelligent vision applications. Despite recent efforts in this task, it is still challenging to deal with novel privacy attributes and novel privacy attack models that are unavailable during the training phase. In this paper, from the perspective of meta-learning (learning to learn), we propose a novel Meta Privacy-Preserving Action Recognition (MPPAR) framework to improve both generalization abilities above (i.e., generalize to *novel privacy attributes* and *novel privacy attack models*) in a unified manner. Concretely, we simulate train/test task shifts by constructing disjoint support/query sets w.r.t. privacy attributes or attack models. Then, a virtual training and testing scheme is applied based on support/query sets to provide feedback to optimize the model's learning towards better generalization. Extensive experiments demonstrate the effectiveness and generalization of the proposed framework compared to state-of-the-arts.

## 1 Introduction

Recently, smart home cameras such as Amazon Echo and Google Nest Cam, have been widely used in millions of families to provide intelligent monitoring services to enhance security (e.g., detecting unusual behaviors and alerting householders remotely) [1, 2]. However, the widespread use of such cameras has raised privacy concerns and even pressured changes in the regulations/laws in Europe and US [13], as it generally requires uploading device-captured visual data that often contains rich privacy information (such as face, gender and skin color, etc) to the intelligent system deployed on cloud or public servers for analysis, which leads to the risk of privacy leakage [3, 4]. While traditional cryptographic solutions can provide video encryption during transmission between the local camera and the intelligent system, they would struggle with preventing authorized agents (e.g., the system administrator) from misusing the privacy information [13, 14].

Therefore, there is an urgent need to find an appropriate *anonymization transformation* to erase privacy information from the captured raw visual data at the local camera end (before the upload), while still enabling certain target tasks required by the intelligent system [5, 6, 7]. Meanwhile, since action recognition is a fundamental video understanding task with wide applications (such as the above-mentioned smart home cameras), there is a growing interest in studying anonymization transformation for privacy preservation in video action recognition, i.e., Privacy-Preserving Action

---

[*]CorrespondingAuthor

37th Conference on Neural Information Processing Systems (NeurIPS 2023).

Recognition (PPAR) [8, 9, 10, 11, 12, 13, 14, 15]. The goal of PPAR is to train an anonymization model to transform the captured raw videos, so that the transformed (anonymous) videos not only avoid undesired privacy disclosure but also enable performing the specified action recognition task.

Not that while using skeleton-based models could naturally handle the privacy concerns in action recognition, the research regarding privacy preservation in RGB-based models still remains important and practical. Here are the reasons: In many scenarios, it is sometimes also necessary to incorporate information beyond action into the action recognition model. For instance, in the kitchen scenario [18] which contains many fine-grained actions, there are many actions that can be similar, such as wash dishes and cut vegetables (both require one hand to hold onto the object while the other hand performs periodic shaking motions). In this case, accurately recognizing the action requires richer contextual information from the background and surroundings. In such a scenario, simply using the skeleton to remove all other information will lead to performance degradation [19]. This indicates that in many scenarios, we still require RGB videos to provide reliable action recognition. Therefore, In RGB-based action recogtion, learning how to remove privacy information while retaining both action information and related visual cues for accurate action recognition becomes extremely important. Recently, it has attracted increasing attention and a series of works [13, 15, 28] has been proposed, reflecting the significance of tackling privacy concerns within RGB-based action recognition. Furthermore, RGB-based action recognition has already been extensively deployed in many household applications [20], such as Amazon Echo and Google Nest Cam. Therefore, the privacy preservation for RGB-based action recognition (i.e., PPAR) is of utmost urgency.

In PPAR, some video downsampling-based methods [8, 9, 10] propose to produce extremely low-resolution videos to create privacy-preserving "anonymous videos". Besides, there exist some obfuscation-based methods [11, 12] that propose to produce "anonymous videos" by blurring videos. Although these video processing-based methods have shown promising results, simply discarding visual content without end-to-end learning often cannot provide a good trade-off between action recognition and privacy preservation. To handle this issue, recent methods [13, 14, 15] propose to explicitly optimize the action-privacy trade-off by training an anonymization model to remove privacy information through an adversarial learning framework. Specifically, the adversarial learning framework incorporates the action recognition model and the privacy classification model, to train the *anonymization model* using a minimax optimization strategy where the action recognition loss is minimized while the privacy classification loss is maximized, ultimately achieving a good trade-off between action recognition and privacy preservation. However, despite the progress, these methods still struggle in real-world applications due to their limited generalization ability, which can manifest in the following two aspects:

– In real-world applications, video data often contains very rich personal information that are difficult to be exhaustively enumerated, such as palm prints, fingerprints, faces, credit cards, cellphone screen information, etc. This makes it almost impossible to ensure that all potential privacy information has been comprehensively labeled in the training data. Existing works focusing on locating and removing privacy attributes through labeled supervision do not explicitly deal with the *novel (unseen) privacy attributes*, which can be unsuitable to handle the real-world applications where the video data can include a broad range of potential privacy attributes.

– On the other hand, an ideal privacy-preserving framework should also be model-agnostic, i.e., preventing various possible privacy attack (i.e., privacy classification) models from stealing privacy information [13, 14]. However, previous studies often use specified privacy attack models for privacy-preserving training, which might limit their privacy-preserving performance when facing *novel privacy attack models*. Furthermore, due to the rapid development and evolution of privacy classification models [16, 17], it is impractical to collect all possible privacy attack models for privacy-preservation training to ensure the attack model during testing is always seen before.

Thus in this paper, we aim to build a privacy-preserving action recognition framework with better generalization abilities in terms of both the above-mentioned aspects: handling *novel privacy attributes* as well as *novel privacy attack models*. We observe that these two aspects both require the anonymization model to learn generalizable knowledge about identifying and removing privacy information, that can generalize beyond the given dataset (with limited labeled privacy attributes) and specified attack models. Despite the conceptual simplicity, how to guide anonymization models to learn such generalizable knowledge is challenging. Here from the perspective of meta learn-

ing, we propose a novel framework, Meta Privacy-Preserving Action Recognition (**MPPAR**), to simultaneously improve both generalization abilities of the anonymization model.

Meta learning, also known as learning to learn, aims to enhance the model generalization capability by performing virtual testing during model training [21, 22]. Inspired by this, to improve the generalization capability of the anonymization model, our framework incorporates a *virtual training and testing* scheme that consists of three steps: virtual training, virtual testing and meta optimization. Specifically, we first construct a support set for virtual training, and a query set for virtual testing. Since we aim to improve the model generalization capability w.r.t. both *novel privacy attributes* and *novel privacy attack models*, we construct the support set and query set accordingly as follows. For handling *novel privacy attributes*, we split the training data to construct a support set and a query set where the videos of the query set contain labeled privacy attributes that are unseen in the support set. Similarly, for handling *novel privacy attack models*, we split a collection of privacy attack models (for training) into a support set and a query set, which contain different attack models.

Based on the constructed sets, we design a novel *virtual training and testing* scheme. Specifically, we first train the model over the support set (i.e., virtual training), and then test the trained model on the query set (i.e., virtual testing). Based on the testing performance (loss) on the query set, we perform the meta optimization to update the model for better generalization capability. Since the query set contains novel privacy attributes (and novel attack models) w.r.t. the support set, by improving the model's testing performance on the query set after training on the support set via *virtual training and testing*, the model is driven to learn more generalizable knowledge that can help remove potentially unseen privacy attributes (and defend against unknown privacy attackers), during virtual training. In this way, we can effectively improve the model generalization capability, even when handling samples with novel privacy attributes and facing attacks from novel privacy attack models.

## 2 Related Work

**Privacy-Preserving Action Recognition (PPAR).** Existing PPAR methods can be broadly classified into three main categories: (1) downsampling-based methods, (2) obfuscation-based methods, and (3) adversarial-learning-based methods. Generally, downsampling-based methods [8, 9, 10] propose to down-sample each video frame into a low-resolution frame to anonymize the privacy information. Obfuscation-based methods [11, 12] mainly use off-the-shelf object detectors to first detect privacy regions, and then blur the detected regions. Recently, in order to achieve a better action-privacy trade-off, adversarial-learning-based methods [13, 14, 15, 28] adopt a minimax optimization strategy (i.e., adversarial learning) to simultaneously optimize the action recognition and the privacy preservation. Specifically, Sudhakar et al. [15] proposed a BDQ anonymization model to preserve privacy in a spatio-temporal manner. While effective in dealing with seen privacy attributes and attack models, this method may struggle in handling the generalization problem (e.g., *novel attributes* or *novel attack models*). Although Wu et al. [13, 14] have proposed a model-ensemble strategy to enhance the model generalization ability to novel attack models, such a strategy might make the trained anonymization model biased towards the attack models used for training. Dave et al. [28] proposed a novel contrastive learning approach to train a visual encoder for attribute-generalizable privacy preservation. Nevertheless, due to its focus on generalization learning without utilizing labeled data, it may lead to sub-optimal privacy protection performance when compared to supervised methods on the labeled attributes. Differently, in this paper, we propose a novel framework that not only retains the benefits from learning with labeled data, but also simultaneously improves the privacy-preserving performance on both *novel privacy attributes* and *novel privacy attack models*, in a unified manner.

**Meta Learning.** The paradigm of learning to learn, known as meta learning, has emerged primarily to tackle the problem of few-shot learning [21, 22, 23, 24]. Specifically, MAML [21] and its subsequent works [22, 23] aim to learn a good initialization of network parameters to achieve fast test-time updates and adapt to new few-shot learning tasks. More recently, meta learning has also been explored in other domains [25, 26, 27] to enhance model performance without requiring test-time updates. Different from typical meta learning methods for few-shot learning, our method aims to address the challenging PPAR problem involving novel attributes and unknown attack models. To this end, we propose a novel framework to optimize the PPAR model via a *virtual training and testing* scheme over the carefully constructed support and query sets.

## 3  Basic Model

Here we first introduce an adversarial-learning-based PPAR model and its training strategy, which has shown promising performance and served as the basis of many state-of-the-art methods [13, 14, 15]. In this paper, we also adopt this model as the basis of our framework. In this basic model, the original video data $X$ is sent into the anonymization model $f^D$ to generate the anonymous video $f^D(X)$. To train the anonymization model $f^D$, its output (i.e., the anonymized video $f^D(X)$) is sent into an action recognition model $f^A$ and a privacy classification (attack) model $f^P$, respectively. The action recognition model predicts the class of the action presented in the anonymized video, while the privacy classification model predicts the existence of each related privacy attribute in the video. The three models ($f^D$, $f^A$, and $f^P$) are all learnable neural networks, and are trained under an adversarial learning algorithm. Specifically, the optimization objective for updating the anonymization model $f^D$ is defined as:

$$\min_{f^D} L, \quad \text{where}: \quad L = L_{action}(f^A(f^D(X)), Y^A) - \gamma L_{privacy}(f^P(f^D(X)), Y^P), \qquad (1)$$

where $Y^A$ denotes the action recognition label, $L_{action}$ denotes the classification (cross-entropy) loss. $Y^P$ represents privacy attribute labels, $L_{privacy}$ is the multiple binary cross-entropy loss for multiple privacy labels and $\gamma$ is a weight coefficient to balance these two terms. By minimizing $L_{action}$ and $-L_{privacy}$ (i.e., maximizing $L_{privacy}$), the anonymization model $f^D$ is optimized to learn to retain the video content that can maintain the action recognition accuracy while removing privacy attribute information as much as possible to lower the privacy classification accuracy, thus to fulfill the goal of privacy-preserving action recognition.

The optimization objective for updating the action recognition model $f^A$ is defined as:

$$\min_{f^A} L_{action}(f^A(f^D(X)), Y^A). \qquad (2)$$

The optimization objective for updating the privacy classification model $f^P$ is defined as:

$$\min_{f^P} L_{privacy}(f^P(f^D(X)), Y^P). \qquad (3)$$

Eq. 2 and 3 are used to maintain the reliable performance of the action recognition model and the privacy classification model, respectively. The total learning algorithm is handled as a competition (adversarial) game [13, 14] among Eq. 1, 2 and 3. Thus after competition with each other, the capabilities of all three models (i.e., $f^D$, $f^A$ and $f^P$) can be gradually enhanced.

## 4  Proposed Framework

In PPAR, existing state-of-the-art models: (1) generally focus on removing privacy information based on given privacy attribute labels seen by the model during training, and (2) typically handle the privacy preservation task by fooling the predefined privacy classification (attack) models. As a result, after training with the given attribute labels and attack models, these methods often struggle when encountering new privacy attributes or new attack models that are unseen during training. To handle this issue, we aim to guide the model to learn more generalizable knowledge that can be applied for privacy preservation in an attribute-agnostic and model-agnostic manner,

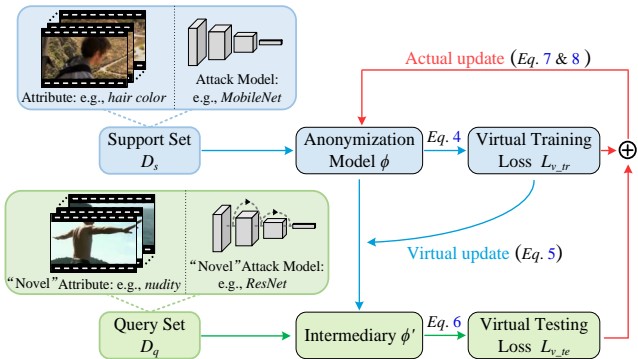

Figure 1: Illustration of our virtual training and testing scheme. This scheme is handled by three steps: Virtual Training (marked in blue), Virtual Testing (green), and Meta Optimization (red).

which however is non-trivial. Here from the perspective of meta learning, we propose a novel MPPAR framework to improve generalization abilities of the anonymization model w.r.t. the above two aspects in a unified manner.

As mentioned before, our framework is based on the Basic Model that contains three training objectives for updating $f^D$, $f^A$ and $f^P$, respectively. Since we aim to improve $f^D$ for better generalization, in our framework, we only modify the training of $f^D$ (Eq. 1), while the updates of $f^A$ (Eq. 2) and $f^P$ (Eq. 3) remain the same. More specifically, our framework incorporates a *virtual training and testing* scheme for training $f^D$. In this scheme, taking the procedures for handling *novel privacy attributes* as an instance, we intentionally split the original training set to construct a support set $D_s$, and a query set $D_q$, where $D_q$ contains novel privacy attribute labels w.r.t. $D_s$. Then, we first train the anonymization model $f^D$ using the support set $D_s$ (virtual training), and then test the model's performance on the query set $D_q$ (virtual testing). As the query set contains novel privacy attributes w.r.t. the support set, if the model can still achieve good testing performance on the query set after being trained on the support set, it indicates that the trained model has learned more generalizable knowledge about removing various potential privacy information (including the novel privacy attributes) that are irrelevant to action recognition. With the virtual testing performance as a feedback, we can optimize the virtual training process to drive the model's learning behavior towards learning more attribute-generalizable knowledge. Via similar process, we can also optimize the model to handle *novel privacy attack models*. Below, we first describe the *virtual training and testing* scheme, and then discuss how we construct the support set and query set.

## 4.1 Virtual Training and Testing

— **Virtual Training.** As shown in Fig. 1 (blue), during virtual training, we train the anonymization model via conventional gradient descent with the support set $D_s$. Specifically, we denote the parameters of the anonymization model as $\phi$, and calculate the virtual training loss $L_{v\_tr}$ as:

$$L_{v\_tr}(\phi) = L(\phi, D_s) \tag{4}$$

where $L$ is the loss function (defined in Eq. 1) for updating the anonymization model. After calculating the virtual training loss, we can update our anonymization model's parameters $\phi$ via gradient descent:

$$\phi' = \phi - \alpha \nabla_\phi L_{v\_tr}(\phi) \tag{5}$$

where $\alpha$ is the learning rate for virtual training. In this way, the model may be easy to learn the knowledge specific to the support set $D_s$ (e.g., removing privacy attributes labeled in the support set $D_s$). However, it might not be beneficial for the model performance on the query set (e.g., the query set contains novel privacy attributes). Therefore, in this step, we do not actually update the anonymization model to be $\phi'$ (hence the term "virtual"). Instead, the virtually updated model parameters $\phi'$ merely serve as an intermediary to calculate the virtual testing loss $L_{v\_te}$, which is described in the next step.

— **Virtual Testing.** As shown in Fig. 1 (green), after virtual training, we then evaluate the performance of the *virtually trained* model on the query set $D_q$:

$$L_{v\_te}(\phi') = L(\phi', D_q) \tag{6}$$

As we intentionally make the query set contain novel attributes w.r.t. the support set, this virtual testing loss $L_{v\_te}(\phi')$ can indicate the model's *generalization capability to handle the privacy attributes beyond the support set* after virtual training. The lower the virtual testing loss is, the better the model (after training) generalizes to unseen attributes. Thus, $L_{v\_te}(\phi')$ can serve as a *feedback* to drive the model to adjust its training on the support set towards learning more generalizable knowledge via the following meta optimization step.

— **Meta Optimization.** As discussed above, we aim to optimize the initial model $\phi$, so that after learning (update) on the support set (i.e., $\phi \rightarrow \phi'$), it can also obtain good testing performance (i.e., low $L_{v\_te}(\phi')$) on the query set that contain novel privacy attributes against the training data. To this end, we draw inspirations from MAML [21] and formulate the meta optimization objective as:

$$\begin{aligned} &\min_\phi \ L_{v\_tr}(\phi) + L_{v\_te}(\phi') \\ = &\min_\phi \ L_{v\_tr}(\phi) + L_{v\_te}\big(\phi - \alpha \nabla_\phi L_{v\_tr}(\phi)\big) \end{aligned} \tag{7}$$

where the first term $L_{v\_tr}(\phi)$ indicates the model's training performance, and the second term $L_{v\_te}(\phi')$ represents the model's testing performance (on novel privacy attributes) after virtual updating. Note that instead of optimizing $\phi$ and $\phi'$ sequentially, the meta optimization mentioned

above is performed solely on the initial model $\phi$, while $\phi'$ merely serves as an intermediary for evaluating the model's generalization performance on novel privacy attributes. Intuitively, the objective in Eq. 7 is to learn how to train $\phi$ on the support set (i.e., lower $L_{v\_tr}(\phi)$) for better generalization performance (i.e., lower $L_{v\_te}(\phi')$). Based on Eq. 7, we *actually* update the model with respect to $\phi$ as:

$$\phi \leftarrow \phi - \beta \nabla_\phi \Big( L_{v\_tr}(\phi) + L_{v\_te} \big( \phi - \alpha \nabla_\phi L_{v\_tr}(\phi) \big) \Big) \tag{8}$$

where $\beta$ denotes the learning rate for meta-optimization. The above process of meta optimization is illustrated in Fig. 1 (red). As both $L_{v\_tr}(\cdot)$ and $L_{v\_te}(\cdot)$ in Eq. 8 are based on loss $L(\cdot)$ (Eq. 1), which is proposed to learn to keep the action recognition accuracy while removing privacy attribute information, thus by learning the meta optimization updating rule in Eq. 8, the anonymization model can be guided to learn more generalizable knowledge about retaining action clues while removing other irrelevant visual content that includes various potentially unseen privacy attributes.

Here we provide an intuitive explanation of the meta optimization. In the virtual training step, we simply learn a model on the support set (i.e., $\phi \rightarrow \phi'$). The model $\phi'$ captures knowledge of the provided set well, yet may suffer from the limited generalization ability when handling novel attributes. Hence, it is critical to adjust the training process to optimize the model in a less biased way. To this end, we add a virtual testing loss term as shown in Eq. 7. Since we intentionally design the support set and query set to be disjoint, the virtual testing loss naturally serves as a regularization (feedback) to penalize the learning objective with second-order gradients (i.e., meta gradients, see Eq. 8) when the model generalizes poorly after virtual training. As a result, the biases of the training set are suppressed and the model turns to learn more generalizable knowledge that can help remove as much action-irrelevant information (including various potential privacy attributes) as possible. Note again, while we illustrate the above *virtual training and testing* scheme using *privacy attributes* as an example, this scheme is equally applicable for handling *novel privacy attack models*. We will discuss how to simultaneously address both aspects in Sec. 4.2.

From the above analysis, we show that the efficacy of our framework lies in *incorporating the virtual testing performance to provide the feedback (meta gradients) to optimize the training process*, which enables the model to learn how to train itself to exploit the desired general knowledge for generalization to unseen concepts, such as novel privacy attributes and novel attack models. Note that we do not aim to use the query set to simulate the real testing scenario (which is unknown during training) for model to learn. Instead, we only need the query set to contain novel concepts w.r.t. the support set, so as to provide an effective feedback to drive the model's learning to learn more set-generalizable knowledge and less set-specific knowledge.

## 4.2 Set Construction

Based on the above *virtual training and testing* scheme, we can achieve the two target objectives: generalizing to *novel privacy attributes* and *novel attack models*, in a unified framework by constructing respective support and query sets as follows: Given the training data $X_{train}$ and a set of privacy attack models (for training) $f^P_{train}$, we construct a support set $D_s$ and a query set $D_q$ at the beginning of each training iteration. Specifically, for every odd-numbered iteration, $D_s$ and $D_q$ are constructed by splitting the training data $X_{train}$ to construct two subsets: $X_s$ and $X_q$, where $X_q$ contains novel privacy attributes w.r.t. $X_s$ (i.e., $D_s = X_s$ and $D_q = X_q$ in this case). Thus at each odd-numbered iteration, we train the anonymization model to learn to generalize to *novel privacy attributes* via the *virtual training and testing* scheme based on the constructed $X_s$ and $X_q$. For every even-numbered iteration, $D_s$ and $D_q$ are constructed by splitting the privacy attack models $f^P_{train}$ to construct $f^P_s$ and $f^P_q$ that contain different attack models (i.e., $D_s = f^P_s$ and $D_q = f^P_q$ in this case). Thus at each even-numbered iteration, we train the anonymization model to learn to generalize to *novel attack models* based on $f^P_s$ and $f^P_q$. In this odd-even way, we alternate the learning of attribute-wise and model-wise generalization iteratively, thus enhancing both generalization capabilities of the model. This means that, due to the *unified nature of our scheme*, enhancing the model's both generalization abilities has now been reduced to a straightforward designing of their respective set construction methods. Below, we separately discuss each construction method.

**A. Constructing sets w.r.t. novel privacy attributes.** With respect to the attribute labels, we construct the support set and query set in two steps. **Step (A1)** *At the beginning of each epoch*, we intentionally split the training data $X_{train}$ into two subsets $\{X_1, X_2\}$, where $X_2$ contains data with novel privacy attributes w.r.t. $X_1$. **Step (A2)** *At the start of every odd-numbered training iteration,*

we select a batch of data from the first subset $X_1$ to construct a support set $X_s$, and select a batch of data from the second subset $X_2$ to construct a query set $X_q$.

**B. Constructing sets w.r.t. novel attack models.** In a similar manner, we also construct the support and query sets w.r.t. privacy attack models in the following two steps. **Step (B1)** *At the start of each epoch*, we split all models in $f^P_{train}$ into two subsets $\{f^P_1, f^P_2\}$ with different privacy attack models. **Step (B2)** *Then at the start of every even-numbered training iteration*, we randomly select one model from $f^P_1$ as the support set $f^P_s$ and one model from $f^P_2$ as the query set $f^P_q$.

Note that at each odd-numbered iteration (for attribute-wise generalization learning), we randomly select one privacy attack model from $f^P_{train}$ for anonymization model's training (all models in $f^P_{train}$ are trained by the adversarial learning algorithm). At each even-numbered iteration (for model-wise generalization learning), we randomly sample a batch of data from $X_{train}$ to train the anonymization model. Moreover, to help the anonymization model to cover a wide range of possible attribute shift (and attack model shift) from the provided data set (and model set), instead of fixing $\{X_1, X_2\}$ (and $\{f^P_1, f^P_2\}$) during the whole training process, at the beginning of each training epoch, we re-split the training data $X_{train}$ (and training attack models $f^P_{train}$) to reconstruct $\{X_1, X_2\}$ (and $\{f^P_1, f^P_2\}$). It is worth mentioning that our method is convenient to use, since we only need to construct the corresponding sets via the above strategy and change the model training loss without the need to change model structures.

Notewise, an alternative approach to achieve both generalization objectives in a unified framework is to perform both attribute-wise and model-wise learning simultaneously in each iteration, which is also effective for enhancing model generalization capability w.r.t. the above-mentioned aspects, while our method with odd-even alternation performs slightly better in both training convergence and performance (see Sec. 5.5), since it decouples the twofold generalization problem into two individual ones to solve, and thus eases the model training and obtains better performance.

### 4.3 Overall Training and Testing

During training, we follow previous works [13, 14] to alternatively update the three models ($f^D$, $f^A$ and $f^P$) by using the adversarial training algorithm. At the start of each iteration for updating the anonymization model $f^D$, we first construct the support set and query set w.r.t. *privacy attributes* in odd-numbered iterations and *privacy attack models in* even-numbered iterations following Sec. 4.2. Note that, here we only focus on the iterations of updating $f^D$, i.e., we only count the number as odd or even for iterations of updating $f^D$. After that, the constructed support set and query set are used to train the anonymization model $f^D$ through the *virtual training and testing* scheme as discussed in Sec. 4.1. Hence, we alternatingly deal with the attribute-wise and model-wise generalization problem over iterations, achieving the tackling of both problems. The training for updating $f^A$ and $f^P$ remains the same as previous works [13, 14], detailed in Supplementary. During testing, we follow the evaluation procedures of [13, 14, 28] to evaluate the model's generalization ability to novel privacy attack models and novel privacy attributes. We additionally provide an *algorithm* about the overall training scheme in Supplementary.

## 5 Experiments

Previous PPAR works have proposed three evaluation protocols: evaluation on novel privacy attributes only [28]; evaluation on novel attack models only [13, 14]; evaluation on known privacy attributes and attack models [28, 15], and each protocol has its own experimental setting. Following these works, we also evaluate our framework under these three protocols. Besides, to evaluate model generalization capability more comprehensively, we further introduce a new evaluation protocol: evaluation on both novel privacy attributes and novel privacy attack models together. In each of the four protocols: we are given the training data $X_{train}$ and training (attack) models $f^P_{train}$ for model training; after training, we evaluate the trained anonymization model on the testing data $X_{test}$ with testing (attack) models $f^P_{test}$. The difference between these four protocols lies in whether $X_{train}$ and $X_{test}$ (or $f^P_{train}$ and $f^P_{test}$) contain the same privacy attributes (or attack models).

**Benchmarks.** Following previous works [13, 14, 28], we conduct experiments using two benchmarks. The first benchmark, HMDB51-VISPR, is comprised of HMDB51 [31] dataset and VISPR [30] dataset. HMDB51 [31] is a collection of videos from movies and the Internet. It has 6,849 videos with 51 action categories. VISPR [30] is a collection with a diverse set of personal privacy information. Following [13, 14, 28], we use PA-HMDB [13] that contains 515 video samples with both human action and privacy attribute labels to serve as the testing set of HMDB51-VISPR. The second

**Algorithm 1:** Overall Training Scheme

1 **Given** $X_{train}$ and $f^P_{train}$ for training; $X_{test}$ and $f^P_{test}$ for testing.
2 Initialize $\phi$.
3 **for** *E epochs* **do**
4      Process $X_{train}$ and $f^P_{train}$, following Step (A1) and Step (B1) in Sec. 4.2, respectively.
5      $T = 0$.
6      **for** *iterations I in adversarial training* **do**
7          **if** *I is for updating anonymization model $f^D$* **then**
8              **if** *T is odd* **then**
9                  Construct $X_{v\_tr}$ and $X_{v\_te}$ from $X_{train}$, following Step (A2) in Sec. 4.2.
10              **else**
11                  Construct $f^P_{v\_tr}$ and $f^P_{v\_te}$ from $f^P_{train}$, following Step (B2) in Sec. 4.2.
12              **end**
13              Calculate the virtual training loss $L_{v\_tr}$ on $D_s$ (i.e., $X_{v\_tr}$ or $f^P_{v\_tr}$) using Eq. 4: $L_{v\_tr}(\phi) = L(\phi, D_{v\_tr})$.
14              Calculate an updated version of anonymization model ($\phi'$) using Eq. 5: $\phi' = \phi - \alpha\nabla_\phi L_{v\_tr}(\phi)$.
15              Calculate the virtual testing loss $L_{v\_te}$ on $D_q$ (i.e., $X_{v\_te}$ or $f^P_{v\_te}$) using Eq. 6: $L_{v\_te}(\phi') = L(\phi', D_{v\_te})$.
16              Update using Eq. 8: $\phi \leftarrow \phi - \beta\nabla_\phi\Big(L_{v\_tr}(\phi) + L_{v\_te}\big(\phi - \alpha\nabla_\phi L_{v\_tr}(\phi)\big)\Big)$.
17              $T = T + 1$.
18          **else if** *I is for updating action recognition model $f^A$* **then**
19              Randomly sample a batch of data from $X_{train}$ to update $f^A$ using Eq. 2.
20          **else**
21              Randomly sample a batch of data from $X_{train}$ to update $f^P$ using Eq. 3.
22          **end**
23      **end**
24 **end**

benchmark, UCF101-VISPR, consists of UCF101 [29] dataset and VISPR [30] dataset. UCF101 [29] has 13,320 videos taken from 101 action categories. Following [13, 14, 28], we use the testing sets of UCF101 [29] and VISPR [30] for evaluation in this benchmark. In all protocols, $X_{train}$ (or $X_{test}$) is created by sampling from the training set (or the testing set) of the benchmark.

**Evaluation Method.** In all protocols, we follow previous works [13, 14] to evaluate methods in two folds: (1) action recognition performance – whether the anonymous videos still contain the action clues to enable the action recognition model to maintain satisfactory performance; (2) privacy preservation performance – whether the anonymous video can lead to poor performance of the privacy attack models. As for the evaluation of action recognition, we directly apply the trained $f^D$ and $f^A$ to the testing data set $X_{test}$ (i.e., $f^A(f^D(X_{test}))$) and compute the action classification accuracy: the higher the better. As for the evaluation of privacy preservation, to empirically verify that $f^D$ prohibits reliable privacy prediction (attack) for each model in

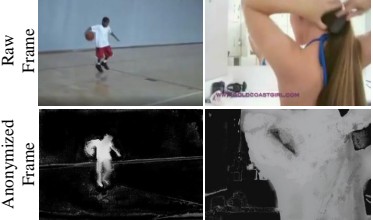

Figure 2: The frames before and after our anonymization. More results are in Supplementary.

the testing model set $f^P_{test}$, we conduct the evaluation as follows: before evaluation, we train each model in $f^P_{test}$ for privacy attribute classification; during evaluation, we apply each trained model in $f^P_{test}$ to perform privacy attribute classification on anonymous videos $f^D(X_{test})$ (anonymized by $f^D$) and compute the classification accuracy. The highest accuracy achieved among all models of $f^P_{test}$ will be by default used to indicate the privacy leakage risk of $f^D$, where a lower value denotes better privacy preservation.

### 5.1 Protocol A: Evaluation on Both Novel Privacy Attributes and Novel Attack Models

**Experimental Setting.** In this evaluation protocol, we follow previous work [28] to set up the training data $X_{train}$ and testing data $X_{test}$, making $X_{train}$ have no overlapped privacy attributes with $X_{test}$. Besides, we follow [13, 14] to set up the training models $f^P_{train}$ and the testing models $f^P_{test}$ with totally different privacy attack models. This means in this protocol, we need to simultaneously handle novel privacy attributes and novel privacy attack models. The detailed list of training/testing attributes and attack models can be found in Supplementary.

Table 1: Results on generalization to both novel privacy attributes and attack models.

| Methods | HMDB51-VISPR | | UCF101-VISPR | |
|---|---|---|---|---|
| | Action (Top-1 ↑) | Privacy (cMAP ↓) | Action (Top-1 ↑) | Privacy (cMAP ↓) |
| Raw Data | 68.5 | 66.3 | 67.4 | 42.4 |
| Downsample 2x [28] | 59.0 | 63.1 | 50.2 | 36.1 |
| Downsample 4x [28] | 52.5 | 61.9 | 43.3 | 32.6 |
| Obf-Blackening [28] | 40.4 | 61.2 | 48.3 | 32.0 |
| Obf-StrongBlur [28] | 41.7 | 59.4 | 49.2 | 29.6 |
| Obf-WeakBlur [28] | 43.8 | 60.8 | 49.5 | 32.5 |
| VITA [13] | 67.9 | 54.5 | 66.5 | 30.2 |
| SPAct [28] | 67.6 | 53.8 | 65.7 | 29.3 |
| Basic Model | 67.3 | 55.2 | 65.6 | 31.7 |
| Ours (full) | **68.1** | **47.4** | **67.0** | **21.8** |

**Implementation Details.** Following [13, 14], we use the Image Transformation model [37] as $f^D$, the action recognition model C3D [32] as $f^A$ and a set of privacy classification models from MobileNet-V2 [33] family as $f^P_{train}$. Since we use C3D [32], we need to split the videos into clips with a fixed frame number. We use the clip with of 16 frames with skip rate of 2. The spatial resolution of each video is resized into $112 \times 112$. For a fair comparison, our method and compared methods use the same model architectures. On each benchmark, we construct the support set with the videos containing 60% of the privacy attributes in the training data $X_{train}$, and use the remaining training data to construct the query set. We set $\gamma$ (in Eq. 1) as 0.4, the learning rate $\alpha$ for virtual training (in Eq. 5) as $5e - 4$, and the learning rate $\beta$ for meta-optimization (in Eq. 8) as $1e - 4$.

**Experimental Results.** As shown in Tab. 1, after applying our *virtual training and testing* scheme to the basic model, our framework, i.e., *Ours (full)*, improves the model performance by a large margin, which demonstrates the effectiveness of our framework in generalizing to novel privacy attributes and novel attack models simultaneously. "Raw Data" denotes directly using raw (clean) videos for testing. Moreover, compared to existing methods, *Ours (full)* achieves significantly better privacy performance while keeping the best action performance, demonstrating the superiority of our framework. Qualitative results are shown in Fig. 2.

## 5.2 Protocol B: Evaluation on Novel Privacy Attributes Only

**Experiment Setting.** In this experiment, we follow the evaluation protocol in [28], which focuses on the generalization to novel privacy attributes, to set up $X_{train}$ and $X_{test}$ containing different attributes, and use the same privacy classification model for both training and testing.

**Implementation Details.** Following [28], we utilize UNet [34] as $f^D$, R3D-18 [36] as $f^A$, and ResNet-50 [35] as $f^P$. Other implementation details remain the same as in Protocol A.

**Experimental Results.** For a fair comparison, our method removes the *virtual training and testing* iter-

Table 2: Results on generalization to novel privacy attributes.

| Methods | HMDB51-VISPR | | UCF101-VISPR | |
|---|---|---|---|---|
| | Action (Top-1 ↑) | Privacy (cMAP ↓) | Action (Top-1 ↑) | Privacy (cMAP ↓) |
| Raw Data | 47.8 | 61.7 | 62.9 | 58.3 |
| Downsample 2x [28] | 38.5 | 58.8 | 54.1 | 52.2 |
| Downsample 4x [28] | 32.4 | 58.2 | 39.7 | 41.5 |
| Obf-Blackening [28] | 20.7 | 57.0 | 53.1 | 53.6 |
| Obf-StrongBlur [28] | 21.3 | 56.9 | 55.6 | 53.7 |
| Obf-WeakBlur [28] | 23.5 | 57.3 | 61.5 | 55.8 |
| VITA [13] | 45.1 | 53.4 | 62.1 | 49.6 |
| SPAct [28] | 44.7 | 45.3 | 62.0 | 47.1 |
| BDQ [15] | 46.6 | 52.2 | 62.3 | 48.8 |
| Basic Model | 44.5 | 54.0 | 61.8 | 50.1 |
| Ours (only attribute) | 47.2 | 43.5 | 62.5 | 32.2 |
| Ours (full) | **47.5** | **43.2** | **62.6** | **32.0** |

ations w.r.t. model-wise generalization and only carries out the attribute-wise iterations, which is denoted as *Ours (only attribute)* in Tab. 2. Compared to other methods, *Ours (only attribute)* brings obvious performance improvement, demonstrating our method's effectiveness in tackling privacy-attribute generalization problem individually. *Ours (full)* denotes the model trained under Protocol A, and here we also directly evaluate its performance under the current protocol. As shown, this model also achieves remarkable trade-offs, demonstrating our method trained for both generalizations can also well handle the generalization scenario where only the attributes are novel.

## 5.3 Protocol C: Evaluation on Novel Privacy Attack Models Only

**Experiment Setting.** We follow the protocol used in [13, 14], which focuses on the generalization to novel privacy attack models, to set up the training and testing models $f^P_{train}$ and $f^P_{test}$ containing totally different privacy attack models. The training data $X_{train}$ and testing data $X_{test}$ contain the same privacy attributes.

**Implementation Details.** For a fair comparison, we use the same model architectures as [13, 14] (Sec. 5.1). Other implementation details are same as Protocol A.

**Experimental Results.** For a fair comparison, in this experiment, our method only adopts *virtual training*

Table 3: Results on generalization to novel privacy attack models.

| Methods | HMDB51-VISPR | | UCF101-VISPR | |
|---|---|---|---|---|
| | Action (Top-1 ↑) | Privacy (cMAP ↓) | Action (Top-1 ↑) | Privacy (cMAP ↓) |
| Raw Data | 67.9 | 76.5 | 67.4 | 46.3 |
| Downsample 2x [28] | 58.9 | 73.2 | 50.2 | 38.3 |
| Downsample 4x [28] | 52.6 | 72.9 | 43.3 | 34.7 |
| Obf-Blackening [28] | 40.2 | 70.4 | 48.3 | 31.1 |
| Obf-StrongBlur [28] | 41.8 | 71.6 | 49.2 | 30.8 |
| Obf-WeakBlur [28] | 43.3 | 71.7 | 49.5 | 35.2 |
| VITA [13] | 66.6 | 64.8 | 66.5 | 34.2 |
| SPAct [28] | 66.3 | 66.7 | 65.7 | 35.0 |
| Basic Model | 66.1 | 66.9 | 65.2 | 35.3 |
| Ours (only model) | 67.3 | 53.9 | 66.9 | 27.7 |
| Ours (full) | **67.4** | **53.6** | **67.0** | **27.3** |

*and testing* iterations w.r.t. model-wise generalization, which is denoted as *Ours (only model)*. As shown in Tab. 3, compared to existing methods and the basic model, *Ours (only model)* can reduce the privacy leakage by a large margin, showing our method's effectiveness in model-wise generalization. Furthermore, *Ours (full)* can also outperform existing methods on the privacy performance while maintaining the best action performance, demonstrating our framework trained for both generalizations can also well handle the generalization where only the attack models are novel.

## 5.4 Protocol D: Evaluation on Known Privacy Attributes and Attack Models

**Experiment Setting.** We follow the protocol used in [28], which also proposes to conduct evaluation on known privacy attributes and privacy attack models, to use $X_{train}$ and $X_{test}$ that contain the same attributes and adopt the same privacy classification model for both training and testing.

**Implementation Details.** For a fair comparison, we follow the previous work [28] using the same model architectures (see Sec. 5.2). Other implementation details are the same as in Protocol A.

**Experimental Results.** In this protocol, since we only have one privacy classification model, we cannot

Table 4: Results on generalization to known privacy attributes and privacy attack models.

| Methods | HMDB51-VISPR | | UCF101-VISPR | |
|---|---|---|---|---|
| | Action (Top-1 ↑) | Privacy (cMAP ↓) | Action (Top-1 ↑) | Privacy (cMAP ↓) |
| Raw Data | 43.8 | 70.6 | 62.9 | 64.6 |
| Downsample 2x [28] | 36.1 | 61.2 | 54.1 | 57.2 |
| Downsample 4x [28] | 25.8 | **41.4** | 39.7 | **50.1** |
| Obf-Blackening [28] | 34.2 | 63.8 | 53.1 | 56.4 |
| Obf-StrongBlur [28] | 36.4 | 64.4 | 55.6 | 55.9 |
| Obf-WeakBlur [28] | 41.7 | 69.4 | 61.5 | 63.5 |
| VITA [13] | 42.3 | 62.3 | 62.1 | 55.3 |
| SPAct [28] | 43.1 | 62.7 | 62.0 | 57.4 |
| BDQ [15] | 43.3 | 61.8 | 62.3 | 53.5 |
| Basic Model | 42.1 | 62.8 | 62.0 | 55.7 |
| Ours (only attribute) | 43.4 | 61.0 | 62.5 | 53.2 |
| Ours (full) | **43.5** | 61.2 | **62.6** | 53.2 |

perform the *virtual training and testing* iterations w.r.t. model-wise generalization, but we can still carry out attribute-wise iterations, as $X_{train}$ contains multiple privacy attributes. Therefore, in this protocol, we adopt *Ours (only attribute)* to fairly compare with others. As shown in Tab. 4, the downsampling-based methods bring obvious performance drop in action recognition. In contrast, our method achieves the closest action recognition performance to *Raw Data*, and meanwhile outperforms other learning-based methods [13, 28, 15] in privacy preservation. This can be attributed to that by training the model to capture more general knowledge, our framework can help the model to better understand the given video and find its privacy information, which thus can generally improve model performance. We can also see that *Ours (full)* shows comparable results with *Ours (only attribute)*.

## 5.5 Ablation Studies

**Impact of alternate learning with odd and even numbered iterations.** To enhance model generalization abilities to both novel privacy attributes and novel privacy attack models, we perform the attribute-wise *virtual training and testing* and model-wise *virtual training and testing* for odd-numbered iterations and even-numbered iterations, respectively. In this alternate way, the model can learn to improve both abilities in the whole training process, simultaneously. To investigate the impact of *alternate learning*, we evaluate the variant by using both *virtual training and testing* strategies at each single iteration, denoted as *attri+model learning*. As shown in Tab. 5 and Fig. 3, *alternate learning* outperforms the variant and shows faster convergence, which demonstrates its effectiveness. **More ablation studies are in Supplementary**

Table 5: Ablation results of attri+model learning and alternate learning.

| Methods | HMDB51-VISPR | | UCF101-VISPR | |
|---|---|---|---|---|
| | Action ↑ | Privacy ↓ | Action ↑ | Privacy ↓ |
| attri+model learning | 67.9 | 47.9 | 66.8 | 22.4 |
| alternate learning | **68.1** | **47.4** | **67.0** | **21.8** |

Figure 3: The loss curves of attri+model learning and alternate learning.

## 6 Conclusion

In this paper, we have proposed a unified framework that improves the generalization ability of anonymization model, through simultaneously improving its performance for handling novel privacy attributes and novel privacy attack models. By carefully constructing the support set and query set with different attributes and attack models, our framework trains the anonymization model with a virtual training and testing scheme and guides it to learn knowledge that is general to scenarios where the privacy attributes and attack models are novel. Extensive experiments have shown the efficacy of our framework under different settings.

## 7 Acknowledgments

This work was supported in part by the National Key Research and Development Program of China under Grant 2022YFA1004100, National Research Foundation, Singapore under its AI Singapore Programme (AISG Award No: AISG2-PhD-2022-01-027), Singapore Ministry of Education (MOE)

AcRF Tier 2 under Grant MOE-T2EP20222-0009, National Research Foundation Singapore through AI Singapore Programme under Grant AISG-100E-2023-121, and SUTD SKI Project under Grant SKI 2021_02_06.

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
