# Joint Attribute and Model Generalization Learning for Privacy-Preserving Action Recognition (Supplementary)

## 1 Additional Ablation Studies

**Impact of constructing attribute-different (and model-different) support set and query set.** In our experiments, we construct the support set and query set to have different privacy attributes and privacy attack models between them in each iteration of our virtual training and testing scheme. Here we assess the variant (*Random Set Construction*) by randomly selecting samples to construct the support set and query set in each iteration. As shown in Tab. 1, our framework (*Attribute/Model-Different Set Construction*) achieves better performance than this variant, which demonstrates the effectiveness of constructing the support set and query set to have different attributes and models.

Table 1: Ablation studies on the effect of constructing support and query sets to have different privacy attributes and privacy attack models. Experiments are conducted under protocol A.

| Methods | HMDB51-VISPR | | UCF101-VISPR | |
|---|---|---|---|---|
| | Action $\uparrow$ | Privacy $\downarrow$ | Action $\uparrow$ | Privacy $\downarrow$ |
| Basic Model | 67.3 | 55.2 | 65.6 | 31.7 |
| Random Set Construction | 67.6 | 53.9 | 66.0 | 30.3 |
| Attribute/Model-Different Set Construction | **68.1** | **47.4** | **67.0** | **21.8** |

**Impact of meta optimization.** In our framework, we update the anonymization model utilizing the *virtual training and testing* scheme through a meta optimization loss: $L_{v\_tr}(\phi) + L_{v\_te}(\phi - \alpha \nabla_\phi L_{v\_tr}(\phi))$. To investigate the impact of such meta optimization, we compare our method (*Meta Optimization*) with a variant (*Joint Training*) that still constructs the support set and query set in the same way, but directly optimizes the anonymization model through $\nabla_\phi(L_{v\_tr}(\phi) + L_{v\_te}(\phi))$. As shown in Tab. 2, our method consistently outperforms this variant on both benchmarks, which shows effectiveness of the meta optimization in our virtual training and testing scheme.

Table 2: Ablation studies on the effect of meta optimization.

| Methods | HMDB51-VISPR | | UCF101-VISPR | |
|---|---|---|---|---|
| | Action $\uparrow$ | Privacy $\downarrow$ | Action $\uparrow$ | Privacy $\downarrow$ |
| Basic Model | 67.3 | 55.2 | 65.6 | 31.7 |
| Joint Training | 67.3 | 55.1 | 65.7 | 31.7 |
| Meta Optimization | **68.1** | **47.4** | **67.0** | **21.8** |

**Impact of splitting $X_{train}$ and $f_{train}^P$ into different subset proportions.** As mentioned in Sec. 4.2 of our main paper, at the beginning of each epoch, we split the training data $X_{train}$ into two subsets $\{X_1, X_2\}$. Meanwhile, we also split the training attack models $f_{train}^P$ into two subsets $\{f_1^P, f_2^P\}$. In this paper, we make the first subset $X_1$ (or $f_1^P$) contain around 60% privacy attributes (or attack models), while the second subset $X_2$ (or $f_2^P$) contains the remaining 40% attributes (or attack models).

37th Conference on Neural Information Processing Systems (NeurIPS 2023).

Here we test two variants. One variant uses 50% attributes (or attack models) for the first subset, and the remaining 50% for the second subset. While another variant uses 70% attributes (or attack models) for the first subset, and the remaining 30% for the second subset. As shown in Tab. 3, our method and these two variants all achieves a better performance than the basic model, showing the robustness of our framework w.r.t. varying subset proportions.

Table 3: Ablation studies on the use of different subset proportions when splitting $X_{train}$ and $f_{train}^P$.

| Methods | HMDB51-VISPR | | UCF101-VISPR | |
|---|---|---|---|---|
| | Action ↑ | Privacy ↓ | Action ↑ | Privacy ↓ |
| Basic Model | 67.3 | 55.2 | 65.6 | 31.7 |
| 50% for the first subset | 68.0 | 47.5 | 66.9 | 22.0 |
| 60% for the first subset | 68.1 | 47.4 | 67.0 | 21.8 |
| 70% for the first subset | 68.1 | 47.6 | 66.8 | 21.9 |

**Training Time.** We evaluate the training time of our framework that trains the basic model with the *virtual training and testing* scheme, and compare it to the training time of the basic model that trains the same network in the conventional training manner without virtual training and testing, on HMDB51-VISPR benchmark, as shown in Tab. 4. We conduct our experiments on an RTX-3090 GPU. Though our method achieves a significant performance gain, it only brings relatively little increase of the training time. Since our approach does not change the anonymization model's structure, our method performs inference almost the same as the basic model.

Table 4: Comparison of training time. Note that our method achieves significantly better performance than the basic model (see Tab. 1 in our main paper).

| Methods | Training Time | Inference Time |
|---|---|---|
| Basic Model | 1.4 days | 37.06 ms |
| Ours | 1.8 days | 37.11 ms |

**Parameter Analysis.** For parameter $\gamma$, which is the weight coefficient of loss $L$ (see Eq. 1 in our main paper), we follow previous works [1, 2] to set its value. In this paper, we only need to study the impact of $\alpha$ (the learning rate in virtual training) and $\beta$ (the learning rate in meta optimization). We present the ablation results of $\alpha$ and $\beta$ in Fig. 1. We can see that the best option of $\alpha$ and $\beta$ are $\alpha = 5e-4$ and $\beta = 1e-4$, respectively.

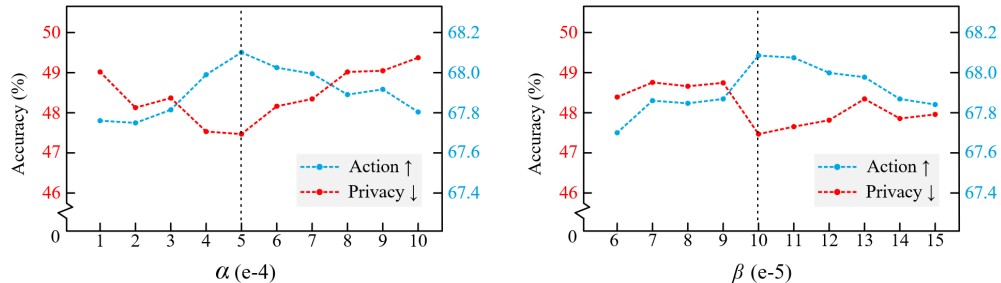

Figure 1: Parameter analysis on $\alpha$ and $\beta$. As for action accuracy, the higher the better. As for privacy accuracy, the lower the better.

## 2 Qualitative Results.

In Fig. 2, we present some qualitative results for generalization to novel attributes and novel privacy attack models. As shown, our method successfully filters out privacy attributes that are seen during training, such as *skin color* and *nudity*, while still enabling the recognition of action clues. Compared to other methods, our method achieves the closest action performance to the raw data. Notably, our method can also effectively remove novel privacy attributes (that are unknown during training), such as *Face* and *Gender*, even using novel privacy attack models cannot detect (identify) these privacy attributes (as denoted by undetected in Fig. 2), which demonstrates our framework can effectively handle various potential privacy attributes and defend against unknown privacy attack models.

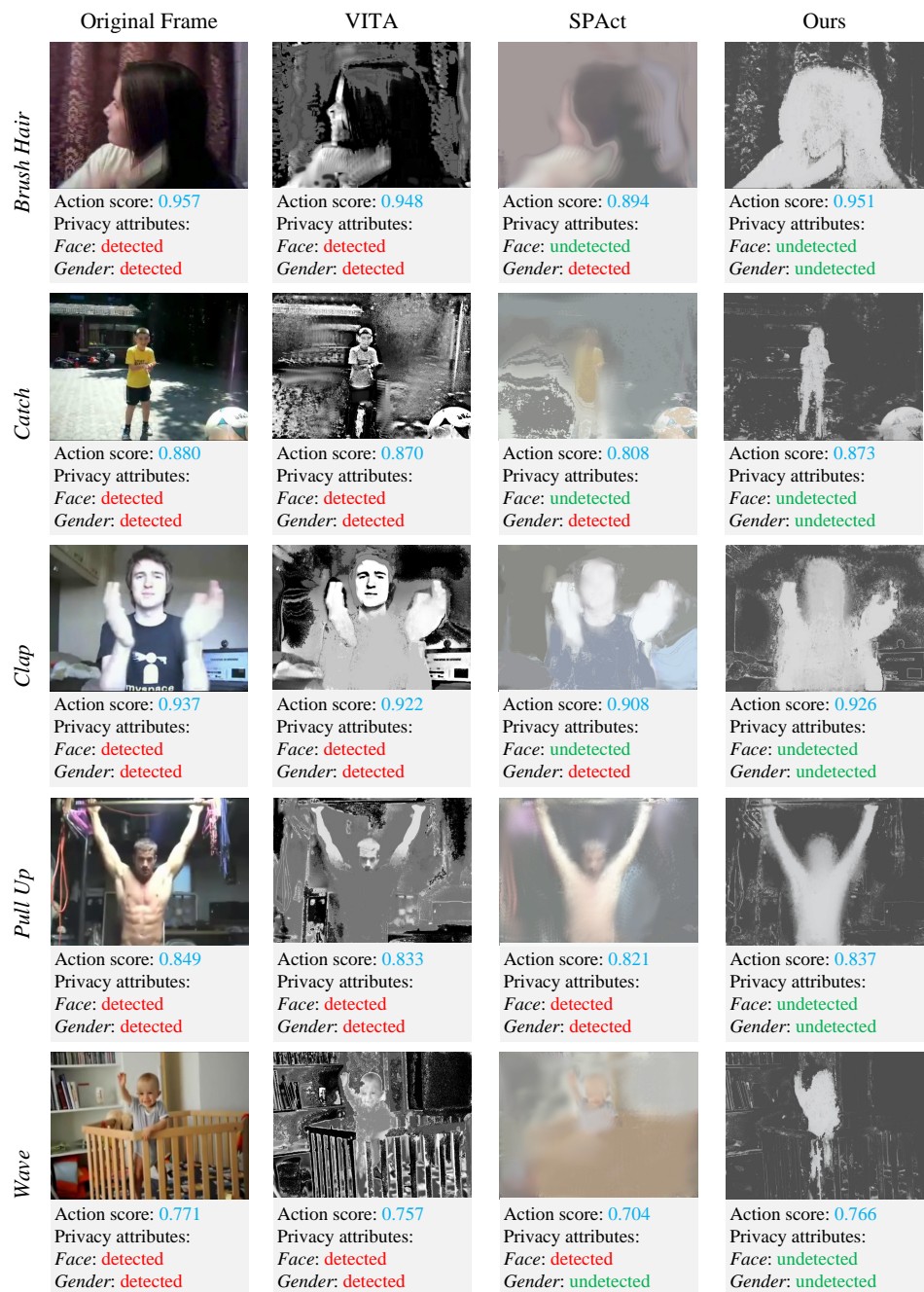

Figure 2: Qualitative results of existing methods and ours on generalization to both novel privacy attributes and novel privacy attack models. For brevity, we only show the center frame of each anonymized video. *Face* and *Gender* are novel attributes that are unknown during training. We use detected and undetected to denote if the attribute is detected (identified) by the novel privacy attack models, i.e., undetected represents the privacy attribute is protected.

## 3    Discussion of Our Meta Framework and Train-Validation

In our proposed framework, we first train the model using the support set (i.e., virtual training), and then evaluate the model performance on the query set (i.e., virtual testing). Then the model evaluation performance on the query set is utilized to provide a generalization regularization (feedback) to drive the model training towards learning more generalizable knowledge.

Meanwhile, in a classical train-validation scheme that is often used for hyperparameter selection, we often train the model over the training set, and then evaluate and observe the model performance on the validation set. From this perspective, our meta framework shares some similarities in concepts with the train-validation scheme, but it is worth noting that, our meta framework is totally different from the classical train-validation scheme. We explain the two reasons in more details below.

Firstly, the classical train-validation scheme is often used to adjust the model parameters indirectly through adjusting the hyperparameters, which lacks a mechanism to directly and automatically optimizes the model parameters. While our framework incorporates a feedback mechanism to directly optimize the anonymization model parameters during training to drive the anonymization model to automatically learn more generalizable knowledge. This cannot be done by simply utilizing the train-validation scheme.

Secondly, we intentionally design attribute (and model) differences between the constructed support and query sets in our framework. Hence, utilizing the evaluation result computed on the query set with different attributes (and models) from the virtual training set as the feedback, our framework further encourages the knowledge learned by the anonymization model during training to be more attribute-generalizable and model-generalizable.

Due to the effectiveness of our meta framework with the above designs, our framework achieves superior performance on the evaluated benchmarks.

## 4 More Implementation Details

**The list of training/testing attributes and attack models.** To study the generalization to novel privacy attributes and novel privacy attack models, we conduct experiments using training and testing sets with different attributes and attack models. Here we present the detailed list of training/testing attributes and attack models. As for privacy attributes, following previous work [3], in UCF101-VISPR, we use the data that contains 7 privacy attributes (i.e., color, gender, complete face, partial face, semi-nudity, personal relationships, social relationships) for training, and use the data that contains other 7 privacy attributes (i.e., hair color, race, sports, age, weight, landmark, tattoo) for testing. In HMDB51-VISPR, the training data contains 3 privacy attributes (i.e., skin color, nudity, and personal relationship) and the testing data contains other 2 privacy attributes (i.e., gender and face). The label of each privacy attribute is a binary label, where 0 indicates absence of the attribute and 1 indicates presence of the attribute. As for privacy attack models, we follow previous works [1, 2] using 8 privacy attack (classification) models chosen from MobileNet-V2 [4] family with different width-multiplier parameters for training, and 10 different state-of-the-art privacy attack (classification) models (i.e., ResNet-V1-$\{50, 101\}$ [5], ResNet-V2-$\{50, 101\}$ [6], Inception-V1 [7], Inception-V2 [8], and MobileNet-V1-$\{0.25, 0.5, 0.75, 1\}$ [9]) for testing. A visual sample can have multiple privacy attributes, hence these privacy attack models are multi-binary classification models.

**Training Algorithm.** Algorithm 1 outlines the overall training algorithm of our framework. The thresholds $T_1$ and $T_2$ are set as $T_1 = 70\%$ and $T_2 = 95\%$ following previous works [1, 2]. The training algorithm of the basic model is to replace the steps for updating the anonymization model $f^D$ (i.e., virtual training and testing scheme) into Eq. 1 of the main paper, where the privacy attack model $f^P$ in Eq. 1 is randomly sampled from $f^P_{train}$ at each iteration.

**Details of Set Construction.** As discussed in Sec. 4.2 in the main paper, during set construction w.r.t. novel privacy attributes, we intentionally split the training data $X_{train}$ into two subsets $\{X_1, X_2\}$, where $X_2$ contains data with novel privacy attributes w.r.t. $X_1$. Since a training sample generally contains multiple privacy attributes, it is not straightforward to split the $X_1$ and $X_2$ to have different attributes. To handle this issue, we execute the following steps: **(1)** First, we gather all attribute categories present in $X_{train}$ and compile them into an attribute set called $Attr$. At each epoch, we randomly split $Attr$ into two subsets $\{Attr_1, Attr_2\}$, where $Attr_1$ and $Attr_2$ contain 60% and 40% attributes, respectively. As mentioned before, the label of each privacy attribute is a binary label, where 0 denotes absence of the attribute and 1 indicates presence of the attribute. Thus, each training sample contain a set of labels that are marked with 0 or 1, where 60% of these labels represent attributes from $Attr_1$, and the other 40% labels represents attributes from $Attr_2$. **(2)** We select samples that contain attributes *solely* from $Attr_1$ (i.e., all "1" labels belong to $Attr_1$) and assign them to $X_1$. **(3)** Similarly, we select samples that with attributes *solely* from $Attr_2$ and assign them to $X_2$. **(4)** After these steps, there may still remain samples that contain attributes from both $Attr_1$ and

**Algorithm 1:** Overall Training Scheme of Our Framework

1  **Given** training data $X_{train}$ and training attack models $f_{train}^P$.
2  **for** *epochs* **do**
3      Process $X_{train}$ and $f_{train}^P$, following Step (A1) and Step (B1) in Sec. 4.2 of main paper, respectively.
4      $I = 0$.
5      **for** *iterations* **do**
6          **if** *Action accuracy of* $f^A$ < *threshold* $T_1$ **then**
7              Randomly sample a batch of data from $X_{train}$ to update $f^A$ using Eq. 2 of main paper.
8              *// Avoid too weak action recognition model $f^A$.*
9          **else if** *Privacy accuracy of any attack model* $f_i^P \in f_{train}^P$ < *threshold* $T_2$ **then**
10             Randomly sample a batch of data from $X_{train}$ to update $f_i^P$ using Eq. 3 of main paper.
11             *// Avoid too weak privacy attack models $f_{train}^P$.*
12         **else**
13             *// The next steps are to update anonymization model $f^D$ via virtual training and testing scheme.*
14             **if** *I is odd* **then**
15                 Construct $X_s$ and $X_q$ from $X_{train}$, following Step (A2) in Sec. 4.2 of main paper.
16             **else**
17                 Construct $f_s^P$ and $f_q^P$ from $f_{train}^P$, following Step (B2) in Sec. 4.2 of main paper.
18             **end**
19             Calculate the virtual training loss $L_{v\_tr}$ on $D_s$ (i.e., $X_s$ or $f_s^P$) using Eq. 4 of main paper: $L_{v\_tr}(\phi) = L(\phi, D_s)$.
20             Calculate an updated version of anonymization model ($\phi'$) using Eq. 5 of main paper: $\phi' = \phi - \alpha\nabla_\phi L_{v\_tr}(\phi)$.
21             Calculate the virtual testing loss $L_{v\_te}$ on $D_q$ (i.e., $X_q$ or $f_q^P$) using Eq. 6 of main paper: $L_{v\_te}(\phi') = L(\phi', D_q)$.
22             Update $f^D$ using Eq. 8 of main paper: $\phi \leftarrow \phi - \beta\nabla_\phi\Big(L_{v\_tr}(\phi) + L_{v\_te}\big(\phi - \alpha\nabla_\phi L_{v\_tr}(\phi)\big)\Big)$.
23             $I = I + 1$.
24         **end**
25     **end**
26 **end**

$Attr_2$ (i.e., some "1" labels belong to $Attr_1$ while some "1" labels belong to $Attr_2$). We randomly select 60% of these samples. For each selected sample, we remove its attribute labels from $Attr_2$, i.e., making all labels belonging to $Attr_2$ become NA (Not Available). After label removal, we assign these samples into $X_1$. Although these samples contains visual content w.r.t. attributes from $Attr_2$, actually, they cannot drive the model to learn these $Attr_2$ attributes, due to the lack of supervision (labels). **(5)** For the remaining 40% samples, we remove the attribute labels from $Attr_1$ and assign them to $X_2$. In step **(2)** (or step **(3)**), before assignment to $X_1$ (or $X_2$), we also remove attribute labels from $Attr_2$ (or $Attr_1$). In this way, we can ensure that when the model is trained on $X_1$, it cannot learn attributes labeled in $X_2$, i.e., $X_2$ contains novel attributes w.r.t. $X_1$. After the construction of $X_1$ and $X_2$, the support set $X_s$ and query set $X_q$ are then obtained by sampling from $X_1$ and $X_2$, respectively. In our virtual training and testing scheme, by improving the model generalization performance on the query set $X_q$ after training on the support set $X_s$, the anonymization model is encouraged to learn more attribute-generalizable knowledge.

As for the set construction w.r.t. novel privacy attack models, at each epoch, we split the training attack models $f_{train}^P$ into two subsets $\{f_1^P, f_2^P\}$ with $f_1^P$ containing 60% of models and $f_2^P$ containing 40%. Then, the support set $f_s^P$ and query set $f_q^P$ are obtained by sampling from $f_1^P$ and $f_2^P$, respectively. By learning how to train the anonymization model with $f_s^P$ for better generalization to $f_q^P$, the anonymization model is guided to learn more model-generalizable knowledge.