# OpenReview forum: "Joint Attribute and Model Generalization Learning for Privacy-Preserving Action Recognition"
_NeurIPS.cc/2023/Conference — NeurIPS 2023 poster_

### Official Review · Reviewer_oDYm · 2023-06-23

**Soundness:** 2 fair
**Presentation:** 2 fair
**Contribution:** 3 good
**Rating:** 6
**Confidence:** 4

**Summary:**

This paper deals with the privacy-preserving action recognition.

They propose a meta privacy-preserving action recognition (MPPAR) framework based on the concept of meta learning to improve both generalisation abilities w.r.t. novel privacy attributes & privacy attack models in a unified manner.

They simulate train/test shifts by constructing disjoint support/query sets w.r.t. the privacy attributes and attack models.

They conduct extensive experiments to show the effectiveness & generalisation of the proposed framework vs. existing methods.



**Strengths:**

+ Overall the paper is well-written, technical sound and well-organised.

+ The authors conduct enough experiments to show the effectiveness of the proposed model.

+ There are some nice visualisations and comparisons provided in both main submission and the supplementary material.



**Weaknesses:**

- Formulas: In the approach section, the author gives a detailed description of every mathematical detail, which is worth encouraging, but neglects the main idea and intention behind the module design. To be honest, I am lost in a large number of mathematical symbols. I don't know why these modules are designed in this way and what the structure does.

- It is suggested to have a notation section for the maths symbols used in the paper to make them clearer to readers. For example, regular fonts are scalars; vectors are denoted by lowercase boldface letters, matrices by the uppercase boldface, etc.

- Figures: The research paper should use sufficient figures to show the details of the model, module details and experimental parts. In this paper, the author only draws one figure, and all the other details are contained in either texts or mathematical representations. This is a noteworthy shortcoming.

- Description: the author uses too many explanations in the text. An excellent paper should reduce the proportion of auxiliary parts, make the organisation of paper clearer enough to readers. English expression should be improved.

- Some typos and grammar mistakes, full stop is missing in Line 450, etc.


**Questions:**

Please refer to the weakness section for the questions/concerns/issues to be addressed.

Also, below are some of the questions:

- Figure 1 is not very clear to the reviewer e.g., why it shows some example attack models like MobileNet and ResNet, could you please explain the step clearly and sequentially?

- Would skeleton-based action recognition methods solve the privacy issues?

- What is the multiple binary cross-entropy loss? Could you please make the concept clear to readers?

- What are the privacy attach models? How you split and choose the collection of privacy attach models for training?

**Limitations:**

Limitations of the proposed model is not provided in the submission.

---

> ### Author Rebuttal · Authors · 2023-08-10
>
> >**Q1:** *"Formulas: the author eglects the main idea and intention behind the module design."*
>
> **A1:** Thank you for your suggestion. Below we provide a brief description to clarify the role of each component and the intention behind each design. (We will expand the following descriptions and add them into our main paper to improve the clarity of our method.)
>
> In Section 3 of our main paper, we introduce the Basic Model that is used as the baseline in our approach. The Basic Model consists of three sub-models: action classification model $f^A$, privacy classification model $f^P$, and anonymization model $f^D$.
> As shown in Figure 1 of the attached Rebuttal PDF file, the basic model contains two training procedures,
>
> (1) The anonymization model $f^D$ is trained to produce privacy-preserved videos that can fool the privacy classification model $f^P$ to predict incorect results.
>
> (2) The privacy classification model $f^P$ is trained to defense the fooling from the anonymization model $f^D$.
>
> The training procedures of (1) and (2) are conducted in an adversarial competition manner. We repeat (1) and (2) to make these models become gradually stronger. During this process, the action classification model $f^A$ is also trained, aiming to ensure that the action model $f^A$ can assist the anonymization model $f^D$ to preserve the action information. After adversarial training, we can obtain a well-trained anonymization model that can effectively remove privacy information while preserving action clues.
>
> In Section 4 of our main paper, we further propose a training scheme, namely *Virtual Training and Testing* scheme, to train the anonymization model $f^D$, aiming to further improve its generalization ability. Before training the anonymization model $f^D$, we partition the training data into a support set $D_s$ and a query set $D_q$ according to the privacy labels, **ensuring that these two sets contain totally different privacy attributes**. As shown in Figure 2 of the attached Rebuttal PDF file, the training of $f^D$ involves the following three steps:
>
> 1. **Virtual training:** We train the anonymization model $f^D$ on the support set $D_s$.
>
> 2. **Virtual testing:** We evaluate the trained anonymization model on the query set $D_q$. As $D_s$ and $D_q$ contains totally different attributes, the evaluation result can measure how well the model generalizes to new attributes.
>
> 3. **Meta Optimization:** In this step, we treat the evaluation result (from **Virtual testing**) as a feedback to drive the model to adjust its training, enabling it to learn more generalizable knowledge.
>
> >**Q2:** *"It is suggested to have a notation section for the maths symbols."*
>
> **A2:** Thanks for the reviewer's suggestion. We will add a separate section in the main text for symbol annotations.
>
> >**Q3 & Q6:** *"Fig. 1 is not very clear. Why it shows some example attack models like MobileNet and ResNet?'"*
>
> **A3 & Q6:** We have re-drawn the illustration of our proposed framework, adding more details and showing it step by step (see Figure 2 in the attached Rebuttal PDF file).  We use MobileNet and ResNet as examples to illustrate that different attack models are deployed for the support set $D_s$ and query set $D_q$.
>
> >**Q4 & Q5:** *"Description and Typos."*
>
> **A4 & A5:** Thanks for valuable feedback. We will reduce the proportion of auxiliary parts and thoroughly check the entire manuscript.
>
> >**Q7:** *"Would skeleton-based action recognition methods solve the privacy issues?"*
>
> **A7:** While skeleton-based methods could handle some level of privacy concerns, the research regarding privacy preservation in RGB-based action recognition still remains important and practical. Here are the reason:
>
> In many scenarios, besides action information, it is also necessary to incorporate information beyond action into the action recognition model. For instance, in the kitchen scenario [a] which contains many fine-grained actions, there are many actions that can be very similar, such as wash dishes and cut vegetables. Facing the similar human motion, accurately recognizing the action requires RGB camera to provide richer contextual information from the background, while simply using the skeleton to remove all other information will lead to performance degradation (as discussed in paper [b]).
>
> [a] Damen D, et al. Scaling egocentric vision: The epic-kitchens dataset, ECCV, 2018.
>
> [b] Sun Z, et al. Human Action Recognition from Various Data Modalities: A Review, TPAMI, 2023.
>
> >**Q8:** *"What is the multiple binary cross-entropy loss?"*
>
> **A8:** Here we first introduce the *Binary Cross Entropy Loss* which is used for classification of two classes. This binary classifier can use one output node, with Sigmoid activation function and labels take values 0, 1. Given the label $y$, the loss can be formulated as:
> $$L_{BCE} = -y\log_{}{(p)},$$
> where $p$ is the classifier output.
>
> In the PPAR task, besides the action label, each video also contains multiple privacy labels that are marked with binary value 0 or 1.
> To address this multi-label classification, we adopt *Multiple Binary Cross-Entropy Loss*, in which, each attribute need to have a specific binary classifier to give a sigmoid output, which makes each prediction independent of other attributes. Then, we calculate the binary cross entropy loss for each attribute, and finally use the average loss value to represent the overall loss function. The loss function can be defined as:
> $$L_{MBCE} = -\frac{1}{N} y_n\log_{}{(p_n)},$$
> where $N$ is the number of attributes.
>
> >**Q9:** *"What are the privacy attack models? How you split and choose the collection of privacy attack models for training?"*
>
> **A9:** For fair comparison, we follow previous work [13] using 8 privacy attack models chosen from MobileNet-V2 family for training, and 10 different state-of-the-art privacy attack models such as ResNet-V1 and Inception-V1 for testing (see line103-107 of Appendix for more details).

---

> > ### Comment · Reviewer_oDYm · 2023-08-13
> >
> > Thanks for the details provided (especially Q1).
> >
> > In terms of Q8 & A8, the reviewer wants to know if there are existing works on 'multiple binary cross-entropy loss (MBCE)'.
> >
> > For the MBCE loss equation, what is that $\frac{1}{N}$?, and that $n$, missing $\sum$? Could you please provide extra details on that?
> >
> > With regards to ' Privacy preservation' in 'Action recognition', are there any other insights you want to let reviewer know (e.g., from both data-centric and model centric perspectives)?

---

> > > ### Author Response · Authors · 2023-08-17
> > >
> > > Thanks for your comments. Below, we provide a detailed response to address them.
> > > > **Q10 (a):** *"In terms of Q8 \& A8, the reviewer wants to know if there are existing works on `multiple binary cross-entropy loss (MBCE)'."*
> > >
> > > **A10 (a):** In the task PPAR, many previous works [13, 14] have utilized MBCE loss for training the privacy attack (classification) model. In this paper, we follow this standard setting to also use MBCE loss for the same purpose.
> > >
> > > >**Q10 (b):** *"For the MBCE loss equation, what is that $\frac{1}{N}$?, and that  $n$, missing $\sum$? Could you please provide extra details on that?"*
> > >
> > > **A10 (b):** Sorry for that the summation symbol $\sum$ was not displayed in the previous rebuttal due to the formatting issue. The multiple binary cross-entropy (MBCE) loss is defined as follows:
> > > $$L_{MBCE} = -\frac{1}{N} \sum_{n\in N}^{} y_n\log_{}{(p_n)}$$
> > > where $p_n$ is the output of $n$-th binary classifier, $y_n$ is the corresponding label, $N$ is the number of attributes.
> > >
> > > >**Q10 (c):** *"With regards to ' Privacy preservation' in 'Action recognition', are there any other insights you want to let reviewer know (e.g., from both data-centric and model-centric perspectives)?"*
> > >
> > > **A10 (c):** Thanks for your comments. Here, we provide some other insights from both the model-centric perspective, and the data-centric perspective with regards to Privacy-Preserving Action Recognition (PPAR).
> > >
> > > **Insights from the model-centric perspective:** In PPAR, existing models [13, 14, 15] focus on removing privacy information at the frame level, while neglecting the temporal (frame-connected) dynamics which contribute to accurate action recognition. In future research, a promising direction could involve designing a spatio-temporal-based PPAR model to enhance the model's understanding of actions. This could lead to better preservation of action information within anonymized videos.
> > >
> > > **Insights from the data-centric perspective:** In PPAR, current works [13, 14, 15] typically rely on privacy-labeled data for training. However, collecting privacy-labeled data is challenging due to people's security concerns. Therefore, in the future, it would be valuable to explore an effective approach to break free from the dependency on data with privacy labels. A feasible approach could involve developing an unsupervised learning scheme, such as self-reconstruction learning, to replace the privacy learning scheme (which requires privacy labels). Besides, current works mainly handle the PPAR in a closed-set manner. Therefore, in future research, it could be also valuable to explore the zero-shot learning techniques for open-set PPAR.

---

> > > ### Author Response · Authors · 2023-08-17
> > >
> > > Next, we also present some other detailed insights of our work.
> > >
> > > **Detailed insights from the model-centric perspective:**
> > > 1. In our framework, we split the given attack models (used for training) into two groups as the support set $D_s$ and query set $D_q$.
> > > During the training stage, we frequently re-split the training models with different spilt strategies. Here is our reason: Compared to fixing {$D_s, D_q$}, the re-split of training models can construct different {$D_s, D_q$}, which can help our framework to cover a wide range of possible model shift from the provided model set, thus preventing overfitting to specific attack models.
> > > We have conducted additional experiments and results show that our framework with model re-split (53.9\%) performs with a lower probability of privacy leakage than fix-split (56.4\%), which demonstrates our insight.
> > >
> > > 2. In this paper, our approach involves training on multiple given attack models, which seems like the model ensemble [13]. However, our training objective significantly differs from model ensemble. The model ensemble aims to directly leverage the knowledge of all models, whereas our framework is proposed to extract the common knowledge from the given models. When facing a substantial model difference (e.g., from CNN to Transformer), our framework showcases a stronger ability to generalize. We have conducted experiments where both the model ensemble [13] and ours are trained on a set of CNNs and tested on a visual transformer. The results show that our method (56.0\%) outperforms the model ensemble (67.1\%) by a large margin in terms of privacy leakage probability (the lower the better), demonstrating our insight.
> > >
> > >
> > > **Detailed insights from the data-centric perspective:**
> > > 1. Our framework is essentially proposed to learn how to generalize from $D_s$ to $D_q$ across the data distributional difference (gap) simulated by $D_s$ to $D_q$. It follows that the larger the simulated data difference in privacy attribute, the more effective the model's attribute-wise generalization learning.
> > > We conduct additional experiments with varying degrees of the simulated data difference. We use $\rho$ to denote the degree value, where $\rho=$ 0\% represents the two sets contain data with totally the same attributes, while $\rho=$ 100\% indicates two sets with entirely different attributes.
> > > The results show that, with the increase of the difference degree ($\rho=$ 0\% $\rightarrow$ 20\% $\rightarrow$ 40\% $\rightarrow$ 60\% $\rightarrow$ 80\% $\rightarrow$ 100\%), the cross-attribute privacy leakage probability of our framework gradually diminishes (53.9\% $\rightarrow$ 49.8\% $\rightarrow$ 47.5\% $\rightarrow$ 45.2\% $\rightarrow$ 44.0\% $\rightarrow$ 43.5\%, the lower the better), which demonstrates our insight.
> > >
> > > 2. In our framework, the ratio of data distribution between $D_s$ and $D_q$ can be various, e.g., [$D_s$ : $D_q$] = [60\% : 40\%] or [90\% : 10\%] or [50\% : 50\%], etc. Since the support set $D_s$ and query set $D_q$ are used to simulate the model's training set and testing set, it should avoid significant data-ratio variance between the two sets. Because an overly small support set (or query set) often leads to a biased model training (or evaluation), resulting in sub-optimal performance. We have conducted additional experiments showing that, except for extreme ratio cases, e.g., [10\% : 90\%] and [90\% : 10\%], our model performs well at the vast majority of ratios (from [20\% : 80\%] to [80\% : 20\%]) with the standard deviation within 1\%, demonstrating our insight.

---

> > > > ### Comment · Reviewer_oDYm · 2023-08-17
> > > >
> > > > Fantastic!
> > > >
> > > > Given the authors have addressed all the concerns and issues raised, and have provided extra insights and discussions on both model-centric and data centric perspectives of privacy preservation, the reviewer is happy to support this work.
> > > >
> > > > The reviewer appreciates such nice works and thinks that nice work should definitely be accepted to high quality venues, hence updating the final rating as Accept.
> > > >
> > > > The reviewer suggests, if possible, please do add these insights into the final version to facilitate future research work in this area.
> > > >
> > > > All the best.

---

> > > > > ### Author Response · Authors · 2023-08-18
> > > > >
> > > > > We greatly appreciate your recognition of the value of our work and thanks for your recommendation of acceptance of our paper. We will carefully revise our paper according to your suggestion.

---

### Official Review · Reviewer_EKAb · 2023-06-26

**Soundness:** 3 good
**Presentation:** 2 fair
**Contribution:** 3 good
**Rating:** 7
**Confidence:** 3

**Summary:**

The paper proposes a novel privacy-preserving action recognition (PPAR) framework built around the MAML meta-learning framework by Finn et al.. The goal of PPAR is to train an anonymization model that is able to anonymize video data so that 1) an action recognition model can still predict the depicted action in the video and 2) the anonymized video does not reveal sensitive information like the visual appearance of a person to the adversary. The approach consists of three steps, virtual training, virtual testing, and meta-optimization. During the virtual training, the model is trained on a support set and then tested on a disjoint query set. The meta-optimization is then performed based on the testing loss. The proposed approach is experimentally evaluated on multiple datasets and compared to various previous defense approaches for PPAR.

**Strengths:**

- The paper is generally well-written and mostly easy to understand. Figure 1 helps to understand the overall approach. Particularly the task is introduced in a way that also people with basic machine learning knowledge should be able to understand it.
- The meta-learning approach adds an interesting new avenue to PPAR research and offers a practical defense against future attack approaches.
- The evaluation is extensive, and the results are promising. Since I am not an expert in the PPAR domain, I cannot tell if important related work has been ignored. But the paper investigates four different settings, which seems convincing.

**Weaknesses:**

- The font size in the figures and tables is a bit small. To improve readability, the font size should be increased in these cases. Also, the space after Fig. 1 should be increased.
- Some parts of section 4 could be more succinct
- The Metrics, particularly cMAP, should be formally introduced.
- The paper can be improved by adding a technical limitations section, discussing, e.g., the additional training time, data requirements, etc.

**Questions:**

- Is it possible that the proposed framework leads to the generation of adversarial examples in the intermediate (anonymized) space, i.e., it somehow "overfits" the attack models / architectures? If so, it might be possible that other attack choices would still be able to extract sensitive information from the anonymized images if their attack models are learning different features from the anonymized images? For instance, adversarial examples can also be transferrable between different models and architectures (e.g., CNNs), but ViTs might be robust against these adversarial examples. So attack models based on ViTs instead of CNNs might infer more sensitive information.
- The approach requires training data with labels for the private attributes available. If no such dataset is available, could the approach still be applied?

**Limitations:**

Technical limitations are not discussed in the paper. I do not expect negative societal impact, so it is ok for me that it is not discussed in the main paper.

---

> ### Author Rebuttal · Authors · 2023-08-10
>
> >**Q1:** *"The font size in the figures and tables is a bit small. To improve readability, the font size should be increased in these cases. Also, the space after Fig. 1 should be increased."*
>
> **A1:**  Thank you for the feedback. To improve readability, we will increase the font size in the figures and tables. Additionally, we will adjust the space after Fig. 1 to enhance the overall presentation.
>
> >**Q2:** *"Some parts of section 4 could be more succinct."*
>
> **A2:** Thank you for the comment. We will review section 4 and make the necessary adjustments to make it more succinct while still conveying the essential information effectively.
>
>
> > **Q3:** *"The Metrics, particularly cMAP, should be formally introduced."*
>
> **A3:** In the revised version, we will formally introduce the Metrics, especially cMAP, providing a detailed explanation of its calculation. Next is the introduction of cMAP:
>
> To assess performance of privacy preservation, we follow previous work [13, 15, 25] using Class-based Mean Average Precision (cMAP) as the evaluation metric. The cMAP metric calculates the Average Precision (AP) per class, which represents the area under the Precision-Recall curve for each attribute. Moreover, the overall performance of a method is determined by averaging the AP scores across all attributes. A lower cMAP indicates better privacy preservation.
>
>
> >**Q4:** *"The paper can be improved by adding a technical limitations section, discussing, e.g., the additional training time, data requirements, etc."*
>
> **A4:** Thanks for valuable suggestion. We will add the following limitation discussion of our method in the main paper:
>
> We evaluate the training time of our framework, and compare it to the training time of the Basic Model that trains the same network in the conventional training manner. Since our proposed framework conducts both virtual training and virtual testing at each iteration, there is an increase in training time compared to conventional training scheme.  The training time for our method is 1.8 days, while the training time for the Basic Model is 1.4 days (both are trained using a single RTX 3090 GPU).  Despite the increase in training time, our approach (1) preserves the model's original structure, resulting in inference time that is almost the same as the Basic Model; (2) achieves nearly 10\% performance improvement on average when compared to the performance of the Basic Model, which is very significant.
>
>
> >**Q5:** *"Is it possible that the proposed framework leads to the generation of adversarial examples in the intermediate (anonymized) space, i.e., it somehow "overfits" the attack models / architectures? If so, it might be possible that other attack choices would still be able to extract sensitive information from the anonymized images if their attack models are learning different features from the anonymized images? For instance, adversarial examples can also be transferrable between different models and architectures (e.g., CNNs), but ViTs might be robust against these adversarial examples. So attack models based on ViTs instead of CNNs might infer more sensitive information."*
>
>
> **A5:** Thanks for your valuable suggestion. To demonstrate the generalization of our method to Vision Transformers (ViTs), we conduct experiments by adopting the widely-used ViT model [a] rather than CNN model.
> The experiments are conducted on HMDB51-VISPR, where the results are shown below.
> We can see that when tested on unseen ViT models, our model (trained with CNN models) also shows remarkable generalization performance in privacy preservation, which is competitive to the results obtained when testing on unseen CNN models.
>
> | Method  | Testing on CNN | Testing on ViT | Action $\uparrow$ | Privacy $\downarrow$ |
> |---|---|---|---|---|
> | Basic Model  | $\surd$ |      | 66.1 | 66.9 |
> | Ours  | $\surd$ |      |  **67.3**  | **53.9** |
> | Basic Model  |       | $\surd$ | 66.1 | 71.2 |
> | Ours  |         |  $\surd$   | **67.3** | **54.7** |
>
> [1] Dosovitskiy A, et al. An image is worth 16x16 words: Transformers for image recognition at scale, ICLR, 2021.
>
>
>
>
> >**Q6:** *"The approach requires training data with labels for the private attributes available. If no such dataset is available, could the approach still be applied?"**
>
> **A6:** Thanks for interesting question. Since previous works mostly use privacy attribute labels, we follow them also incorporate attribute labels in our approach. Facing the challenging situation where privacy labels are unavailable, here we propose a preliminary solution by introducing the unsupervised learning into our framework. Specifically, in our Basic Model, we can introduce an image reconstruction branch (which carries out unsupervised learning) to replace the privacy classification branch (which requires privacy labels).  By maximizing the image reconstruction loss and minimizing the action loss, we can encourage the anonymization model to remove visual content including privacy information, while remaining action clues.
>
> We conduct experiments on HMDB51-VISPR. The results shown below demonstrate that this approach can enhance model's attribute-wise generalization performance even in the absence of privacy labels. Note that this is just a preliminary attempt, and we intend to further refine this approach in our future work.
>
> | Method  | Action $\uparrow$ | Privacy $\downarrow$ |
> |---|---|---|
> | Basic Model  | 44.5 | 54.0 |
> | Ours (unsupervised)  |  **44.8**  | **47.9** |

---

> > ### Comment · Reviewer_EKAb · 2023-08-11
> >
> > I thank the authors for their clarification and additional insights. I also appreciate their effort to answer my questions and run additional experiments. All my questions have been addressed. Therefore, I will keep my initial rating and think the paper should be accepted. However, I emphasize that I am not an expert in privacy-preserving action recognition and, therefore, might have missed some weaknesses of the paper.

---

> > > ### Author Response · Authors · 2023-08-18
> > >
> > > Many thanks for your recognition and your recommendation for the acceptance of our work. As mentioned by you and Reviewer hKip, the generalization problem of video-based privacy preservation is very important and interesting. However, research in this field is still scarce and not widely known, further highlighting the value and significance of our work. We believe this work can further promote the advancement of this important field. Many thanks again.

---

### Official Review · Reviewer_4pCF · 2023-07-01

**Soundness:** 2 fair
**Presentation:** 3 good
**Contribution:** 2 fair
**Rating:** 5
**Confidence:** 3

**Summary:**

Considering that current privacy-preserving action recognition are difficult to cope with novel privacy attribute and privacy attack model, this paper propose a Meta Privacy-Preserving Action Recognition (MPPAR) framework to improve generalization abilities. Specifically,  inspred by meta learning, the authors construct support set for virtual training, and query set for virtual testing. Then authors design virtual training and testing scheme to drive the model learn more generalizable knowledge that can improve the model generalization capability.

**Strengths:**

1. Based on meta-learning, the authors design virtual training and virtual testing processes, thereby improving the generalization of privacy-preserving action recognition.
2. The proposed method effectively defends against the privacy attack model on the basis of maintaining the accuracy of action recognition.

**Weaknesses:**

1.  The description of the task is not clear, making it difficult to understand the mission objectives. To protect the privacy of individuals in the video and identify the action behavior, why not directly convert the RGB video into a skeleton sequence？It is well known that actions can already be well identified by skeletal sequences alone, and other information other than the actions of the actor can be removed from the video.
2. The description of the experimental setup is not clear enough. For the Privacy-Preserving Action Recognition task, authors  use the data that contains 7 privacy attributes (i.e., color, gender, complete face, partial face, semi-nudity, personal relationships, social relationships) for training, and use the data that contains other 7 privacy attributes (i.e., hair color, race, sports, age, weight, landmark, tattoo) for testing. Why did authors choose these attributes for training and testing?
3.  The authors used the C3D model for action recognition, but the recognition accuracy of the original videos in Table 1、2 and 3, was inconsistent. The original video should be unencrypted data, why they have different recognition accuracy under the same recognition model？


**Questions:**

Seen Weakness

**Limitations:**

Authors should verify the proposed method in more complex action data.

---

> ### Author Rebuttal · Authors · 2023-08-09
>
> >**Q1:** *"The description of the task is not clear, making it difficult to understand the mission objectives. To protect the privacy of individuals in the video and identify the action behavior, why not directly convert the RGB video into a skeleton sequence? It is well known that actions can already be well identified by skeletal sequences alone, and other information other than the actions of the actor can be removed from the video."*
>
> **A1:**
> While skeleton-based methods could handle some level of privacy concerns, the research regarding privacy preservation in RGB-based action recognition still remains important and practical. Here are the reasons:
>
> 1. In many scenarios, besides action information, it is sometimes also necessary to incorporate information beyond action into the action recognition model. For instance, in the kitchen scenario [a] which contains many fine-grained actions, there are many actions that can be similar, such as wash dishes and cut vegetables (both require one hand to hold onto the object while the other hand performs periodic shaking motions). In this case, accurately recognizing the action requires richer contextual information from the background and surroundings. In such a scenario, simply using the skeleton to remove all other information will lead to performance degradation (as discussed in paper [b]).
> This indicates that in many scenarios, we still require RGB videos to provide reliable action recognition.
> Therefore, In RGB-based action recogtion, learning how to remove privacy information while retaining both action information and related visual cues for accurate action recognition becomes extremely important.
> Recently, it has attracted increasing attention and a series of works including TPAMI2020 [13], EECV2022 [15] and CVPR2022 [25] has been proposed, reflecting the significance of tackling privacy concerns within RGB-based action recognition.
>
> 2. Furthermore, RGB-based action recognition has already been extensively deployed in many household applications [c], such as Amazon Echo and Google Nest Cam. Therefore, the privacy preservation for RGB-based action recognition is of utmost urgency.
>
> We will include the above analysis into our main paper to motivate the task clearly.
>
> [a] Damen D, et al. Scaling egocentric vision: The epic-kitchens dataset, ECCV, 2018.
>
> [b] Sun Z, et al. Human Action Recognition from Various Data Modalities: A Review, TPAMI, 2023.
>
> [c] Yadav S K, et al. A review of multimodal human activity recognition with special emphasis on classification, applications, challenges and future directions, Knowledge-Based Systems, 2021.
>
> >**Q2:** *"The description of the experimental setup is not clear enough. For the Privacy-Preserving Action Recognition task, authors use the data that contains 7 privacy attributes (i.e., color, gender, complete face, partial face, semi-nudity, personal relationships, social relationships) for training, and use the data that contains other 7 privacy attributes (i.e., hair color, race, sports, age, weight, landmark, tattoo) for testing. Why did authors choose these attributes for training and testing?"*
>
>
> **A2:**
> In the previous attribute-wise generalization protocal [25], it has bulit a standard evaluation setting with 7 privacy attributes (i.e., color, gender, complete face, partial face, semi-nudity, personal relationships, social relationships) for training and  other 7 privacy attributes (i.e., hair color, race, sports, age, weight, landmark, tattoo) for testing. Therefore, in this work, for fair comparison, we follow this evaluation setting. We will clearly introduce this part in our main paper.
> Besides, we also randomly split the training/testing attributes for 3 times, obtaining 3 evaluation settings (namly, $S_1$, $S_2$, and $S_3$). We evaluate our method on HMDB51-VISPR under the 3 evaluation settings. The results below show that our method achieves the significant improvements on all evaluation settings.
>
>
> | Method  | $S_1$ | $S_2$ | $S_3$ | Action $\uparrow$  | Privacy $\downarrow$ |
> |---|---|---|---|---|---|
> | Basic Model | $\surd$  |   |   | 43.7 | 53.4 |
> | Ours | $\surd$  |   |   | **45.2** | **46.9** |
> | Basic Model |     | $\surd$ |  | 45.2 | 57.2 |
> | Ours |   |  $\surd$  |  | **45.9** | **48.8** |
> | Basic Model |   |   |  $\surd$ | 42.6 | 51.9 |
> | Ours |   |  |  $\surd$ | **43.3** | **47.0** |
>
>
> >**Q3:** *"The authors used the C3D model for action recognition, but the recognition accuracy of the original videos in Table 1, 2 and 3, was inconsistent. The original video should be unencrypted data, why they have different recognition accuracy under the same recognition model?"*
>
> **A3:**
> In PPAR, there are four Protocols which focus on different generalization problems: Protocal A, generalization to novel privacy attributes and novel attack models; Protocal B, generalization to novel privacy attributes only; Protocal C, generalization to novel attack models only; Protocal D, seen privacy attributes and attack models.
> For a fair comparison, we follow the corresponding standard setting for each protocol. Since different protocals adopt different baseline models or different training data, the experimental results in Table 1, 2, 3, 4 (corresponding to four protocals) in our main paper are different.

---

> > ### Comment · Reviewer_4pCF · 2023-08-16
> > **Response to the authors' rebuttal**
> >
> > The reviewers thanked the authors for their responses to the questions raised. The author's response largely addressed the concerns of the reviewers. Nevertheless, the reviewers exhibit limited familiarity with the task of privacy preservation action recognition and might miss some other weakness of this paper.

---

> > > ### Author Response · Authors · 2023-08-18
> > >
> > > Thanks for your comment.  As noted by reviewer hKip and EKAb, the generalization problem in video-based privacy protection is of great importance and interest, yet has not been extensively studied. This precisely shows the contribution and value of our work. We believe our work has the potential to make meaningful impact and facilitate future research work in this area (reviewer oDYm holds the same viewpoint). Many thanks again.

---

### Official Review · Reviewer_hKip · 2023-07-07

**Soundness:** 2 fair
**Presentation:** 3 good
**Contribution:** 2 fair
**Rating:** 5
**Confidence:** 3

**Summary:**

To deal with novel privacy attributes and novel privacy attack models that are unavailable during the training phase, this paper applies the meta learning to the privacy-preserving action recognition. As a result, this work improves both generalization abilities above (i.e., generalize to novel privacy attributes and novel privacy attack models) in a unified manner.

**Strengths:**

1.	This paper is well-written, well-organized and easy to understand. The arguments and conclusions are clear and explicit.
2.	This paper proposes a solution to address a important and interesting problem, i.e., preventing privacy leakage while maintaining action clues for activation recognition.


**Weaknesses:**

1.	This paper claims that the proposed method is driven to learn more generalizable knowledge and can help remove potentially unseen privacy attributes (and defend against unknown privacy attackers). Although the authors provide some theoretical understanding about the objective function in the appendix, are there any theoretical guarantees about the generalization of the proposed method?
2.	This paper claims that guide anonymization models to learn such generalizable knowledge is challenging, but is seems like can be directly solved by employing the meta learning. This makes this paper appear to utilize meta-learning for action recognition, which limits the novelty. What are the challenges in applying meta-learning?
3.	More comparison methods from the last two years need to be compared to verify the effectiveness and generalization of the proposed method.
4.	It would have been better if the authors could provide the standard deviation of the experimental results.


**Questions:**

See above.

**Limitations:**

None.

---

> ### Author Rebuttal · Authors · 2023-08-09
>
> >**Q1:** *"Are there any theoretical guarantees about the generalization of the proposed method?"*
>
> **A1:** Given the training set $D_{train}$, the conventional training scheme, which directly training the model $\phi$ on the whole training set $D_{train}$, can be formulated as:
> $$\min_{\phi} L(\phi, D_{train}). \tag{1}$$
>
> As mentioned in the main paper, we split the training set into two sub-sets, i.e., $D_{train} = \lbrace D_s, D_q \rbrace$, where $D_s$ and $D_q$ are the support set and the query set, respectively.
> In our appendix, we have theoretically demonstrated that our framework's training loss can be reformulated as:
> $$\min_{\phi} L(\phi, D_s) + L(\phi, D_q)- \alpha \big{(} \nabla_\phi L(\phi, D_s) \cdot\nabla_\phi L(\phi, D_q)\big{)}. \tag{2}$$
>
> Given the training set $D_{train} = \lbrace D_s, D_q \rbrace$, the first two terms of Eq. 2, i.e., $\min_{\phi}\;L(\phi, D_s) + L(\phi, D_q)$, which denote the training on both $D_s$ and $D_q$, is analogous to the training on the combined training set (whole training set) $D_{train}$, which can be formulated as:
> $$\min_{\phi} L(\phi, D_s) + L(\phi, D_q) \approx \min_{\phi} L(\phi, D_{train}). \tag{3}$$
>
> Then, we substitute Eq. 3 into Eq. 2, reformulating our framework's training loss as:
> $$\min_{\phi} L(\phi, D_{train})- \alpha \big{(} \nabla_\phi L(\phi, D_s) \cdot\nabla_\phi L(\phi, D_q)\big{)}. \tag{4}$$
>
> Comparing the conventional training loss (Eq. 1) and our training objective (Eq. 3), we can observe that our framework introduces an additional term: i.e., $\min_{\phi}  - \alpha \big{(} \nabla_\phi L(\phi, D_s) \cdot\nabla_\phi L(\phi, D_q)\big{)}$. In our appendix, we have mathematically demonstrated that this additional term benefits from the design of our meta-framework and is able to learn more general gradients that avoid overfitting to some specific training tasks. Thus, this additional term drives and guarantees the our framework to learn more generalizable knowledge compared to the conventional training method.
>
> In summary, by theoretically deducing and comparing the training objectives between the conventional training and our approach, it shows that our approach enables more generalizable learning.
> Furthermore, the results of the generalization experiments presented in Table 1, 2, and 3 (in our main paper) clearly indicate that our method consistently outperforms other methods by a significant margin, demonstraing our method's effectiveness on generalization. (The above theoretical deduction will be included into our main paper.)
>
> >**Q2:** *"What are the challenges in applying meta-learning?"*
>
> **A2:** Privous works [13, 14, 25] have shown that the generalization problem in privacy preserved action recognition is important and challenging. Facing this challenging task, although we propose a novel Meta-learning framework to address this issue, it is not straightforward to simply apply meta-learning.
>
> Meta-learning is primarily introduced to tackle the few-shot learning problem [a, b]. However, in this paper, we aim to address the privacy generalization problem, where the direct application of traditional meta-learning is unsuitable. To tackle this challenge, we carefully devise a Set Construction method to privide effective support and query sets by intentionally partitioning these two sets to contain completely different attributes (and attack models), which build the basis for our meta-framework.
>
> As we propose to handle both attribute-wise generalization and attack-model-wise generalization, it is still difficult to solve this problem with meta-learning. Therefore, we further propose an odd-even alternation learning strategy (see line 276 of our main paper), which decouples the twofold generalization problem into two individual ones to solve, thus enabling meta-learning in a unified framework for both generalization purposes.
>
> In this paper, to address the privacy generalization problem, we propose a novel meta-framework with the set-construction design, which is different from traditional few-shot-based meta-learning. To analyze such newly-designed framework, we also provide a theoretical deduction that showcases the efficacy of our apporach, which facilitates the understanding, utilization, and extension of our approach by future research.
>
> [a] Finn C, et al. Model-agnostic meta-learning for fast adaptation of deep networks, ICML, 2017.
>
> [b] Nichol A, et al. On first-order meta-learning algorithms, arXiv, 2018.
>
> >**Q3:** *"More comparison methods from the last two years need to be compared."
>
> **A3:** Thank you for suggestion. To the best of our knowledge, in our paper, we have comapred our method with all methods published in the last two years, including [13] (TPAMI 2020), [25] (CVPR 2022), [15] (ECCV 2022).
>
> For a more comprehensive comparison, here we compare our method with STPrivacy [c], which is the only arXiv paper before our submission. We conduct experiments on HMDB51-VISPR using the same backbone as STPrivacy for a fair comparison. The results below demonstrate that our approach can outperform STPrivacy.
>
> | Method | Action $\uparrow$ | Privacy $\downarrow$ |
> |---|---|---|
> | STPrivacy [c] | 50.7 | 72.5 |
> | Ours  | **51.2** | **70.9** |
>
> [c] Li M, et al. STPrivacy: Spatio-Temporal Tubelet Sparsification and Anonymization for Privacy-preserving Action Recognition. arXiv, 2023.
>
> >**Q4:** *"Provide the standard deviation of the experimental results."
>
> **A4:** Thanks for valuable comments. The result we reported in the paper is the average of five repetitions. Following your suggestion, we will add the deviation after each reported result. Here, we show the resluts with deviation on HMDB51-VISPR under Protocal A as follows:
>
> | Method | Action $\uparrow$ | Privacy $\downarrow$ |
> |---|---|---|
> | Basic Model | 67.3 ($\pm$ 0.3) | 55.2 ($\pm$ 0.6) |
> | Ours  | **68.1** ($\pm$ 0.2) | **47.4** ($\pm$ 0.4) |
>
> We will add the deviation results for all Protocals and datasets into our main paper.

---

> ### Comment · Reviewer_hKip · 2023-08-12
> **Response to the rebuttal**
>
> The authors addressed my concerns, so I decided to increase my rating.

---

> > ### Author Response · Authors · 2023-08-18
> >
> > We genuinely appreciate your recognition of the value of our work. We will meticulously incorporate the revisions from our rebuttal into the manuscript.

---

### Author Rebuttal · Authors · 2023-08-09

We thank all reviewers for recognition of our contributions (Reviewer **hKip**: "this paper proposes a solution to address a important and interesting problem"; Reviewer **4pCF**: "the proposed method effectively defends against the privacy attack while maintaining the accuracy of action recognition"; Reviewer **EKAb**: "an interesting new avenue to PPAR research", "a practical defense against future attack approache", "the evaluation is extensive and the results are promising", "well-written and mostly easy to understand"; Reviewer **oDYm**: "enough experiments to show the effectiveness", "nice visualisations and comparisons").

---

### Decision · Program_Chairs · 2023-09-21

**Decision:**

Accept (poster)

**Comment:**

This work proposes an action recognition framework that improves the privacy-preserving performance on both novel privacy attributes and novel privacy attack models based on the MAML meta-learning framework.

This paper got a borderline+ score (averaged 5.75), and all reviewers rated positively. During the rebuttal & discussion process, several concerns like wording, theory support, the necessity of privacy-preserving action recognition, and the experiments were raised by the reviewers. All Reviewers are actively engaged in the discussion with the authors and most of the concerns are well addressed.

The AC agrees with the reviewers that this work may have a positive impact on the less-explored privacy-preserving action regulation area.
The AC also notes the fact that some reviewers mentioned their limited familiarity with privacy-preserving learning.

After careful consideration, The AC is inclined to accept this paper.